# GeNIe: Generative Hard Negative Images Through Diffusion

## Abstract

Data augmentation is crucial in training deep models, preventing them from over-fitting to limited data. Recent advances in generative AI, e.g., diffusion models, have enabled more sophisticated augmentation techniques that produce data resembling natural images. We introduce `GeNIe` a novel augmentation method which leverages a latent diffusion model conditioned on a text prompt to combine two contrasting data points (an image from the source category and a text prompt from the target category) to generate challenging augmentations. To achieve this, we adjust the noise level (equivalently, number of diffusion iterations) to ensure the generated image retains low-level and background features from the source image while representing the target category, resulting in a *hard negative* sample for the source category. We further automate and enhance `GeNIe` by adaptively adjusting the noise level selection on a per image basis (coined as `GeNIe-Ada`), leading to further performance improvements. Our extensive experiments, in both few-shot and long-tail distribution settings, demonstrate the effectiveness of our novel augmentation method and its superior performance over the prior art. Our code is available at: https://anonymous.4open.science/r/GeNIe-F6C6

## 1 Introduction

Augmentation has become an integral part of training deep learning models, particularly when faced with limited training data. For instance, when it comes to image classification with limited number of samples per class, model generalization ability can be significantly hindered. Simple transformations like rotation, cropping, and adjustments in brightness artificially diversify the training set, offering the model a more comprehensive grasp of potential data variations. Hence, augmentation can serve as a practical strategy to boost the model's learning capacity, minimizing the risk of overfitting and facilitating effective knowledge transfer from limited labelled data to real-world scenarios. Various image augmentation methods, encompassing standard transformations, and learning-based approaches have been proposed (Cubuk et al., 2019b;a; Yun et al., 2019; Zhang et al., 2018; Trabucco et al., 2024). Some augmentation strategies combine two images possibly from two different categories to generate a new sample image. The simplest ones in this category are MixUp (Zhang et al., 2018) and CutMix (Yun et al., 2019) where two images are combined in the pixel space. However, the resulting augmentations often do not lie within the manifold of natural images and act as out-of-distribution samples that will not be encountered during testing.

Recently, leveraging generative models for data augmentation has gained an upsurge of attention (Trabucco et al., 2024; Roy et al., 2023; Luzi et al., 2022; He et al., 2022b). These interesting studies, either based on fine-tuning or prompt engineering of diffusion models, are mostly focused on generating *generic augmentations* without considering the impact of other classes and incorporating that information into the generative process for a classification context. We take a different approach to generate challenging augmentations near the decision boundaries of a downstream classifier. Inspired by diffusion-based image editing methods (Meng et al., 2021; Luzi et al., 2022) some of which are previously used for data augmentation, we propose to use conditional latent diffusion models (Rombach et al., 2022) for generating *hard negative* images. Our core idea (coined as `GeNIe`) is to sample source images from various categories and prompt the diffusion model with a contradictory text corresponding to a different target category. We demonstrate that the choice of noise level (or equivalently number of iterations) for the diffusion process plays a pivotal role in generating images that semantically belong to the target category while retaining low-level features

Figure 1: **Generative Hard Negative Images Through Diffusion (`GeNIe`):** generates hard negative images that belong to the target category but are similar to the source image from low-level feature and contextual perspectives. `GeNIe` starts from a source image passing it through a partial noise addition process, and conditioning it on a different target category. By controlling the amount of noise, the reverse latent diffusion process generates images that serve as *hard negatives* for the source category.

from the source image. We argue that these generated samples serve as *hard negatives* (Xuan et al., 2021; Mao et al., 2017) for the source category (or from a dual perspective hard positives for the target category). To further enhance `GeNIe`, we propose an adaptive noise level selection strategy (dubbed as `GeNIe-Ada`) enabling it to adjust noise levels automatically per sample.

To establish the impact of `GeNIe`, we focus on two challenging scenarios: *long-tail* and *few-shot* settings. In real-world applications, data often follows a long-tail distribution, where common scenarios dominate and rare occurrences are underrepresented. For instance, a person jaywalking a highway causes models to struggle with such unusual scenarios. Combating such a bias or lack of sufficient data samples during model training is essential in building robust models for self-driving cars or surveillance systems, to name a few. Same challenge arises in few-shot learning settings where the model has to learn from only a handful of samples. Our extensive quantitative and qualitative experimentation, on a suite of few-shot and long-tail distribution settings, corroborate the effectiveness of the proposed novel augmentation method (`GeNIe`, `GeNIe-Ada`) in generating hard negatives, corroborating its significant impact on categories with a limited number of samples. A high-level sketch of `GeNIe` is illustrated in Fig. 1. Our main contributions are summarized below:

- We introduce `GeNIe`, a novel yet elegantly simple diffusion-based augmentation method to create challenging augmentations in the manifold of natural images. For the first time, to our best knowledge, `GeNIe` achieves this by combining two sources of information (a source image, and a contradictory target prompt) through a noise-level adjustment mechanism.

- We further extend `GeNIe` by automating the noise-level adjustment strategy on a per-sample basis (called `GeNIe-Ada`), to enable generating hard negative samples in the context of image classification, leading also to further performance enhancement.

- To substantiate the impact of `GeNIe`, we present a suit of quantitative and qualitative results including extensive experimentation on two challenging tasks: few-shot and long tail distribution settings corroborating that `GeNIe` (and its extension `GeNIe-Ada`) significantly improve the downstream classification performance.

## 2 RELATED WORK

**Data Augmentations.** Simple flipping, cropping, colour jittering, and blurring are some forms of image augmentations (Shorten & Khoshgoftaar, 2019). These augmentations are commonly adopted in training deep learning models. However, using these data augmentations is not trivial in some domains. For example, using blurring might remove important low-level information from medical images. More advanced approaches, such as MixUp (Zhang et al., 2018) and CutMix (Yun et al., 2019), mix images and their labels accordingly (Hendrycks et al., 2020; Liu et al., 2022; Kim et al., 2020; Cubuk et al., 2020). However, the resulting augmentations are not natural images anymore,

and thus, act as out-of-distribution samples that will not be seen at test time. Another strand of research tailors the augmentation strategy through a learning process to fit the training data (Ding et al., 2024; Cubuk et al., 2019b;a). Unlike the above methods, we propose to utilize pre-trained latent diffusion models to generate hard negatives (in contrast to generic augmentations) through a noise adaptation strategy discussed in Section 3.

**Data Augmentation with Generative Models.** Using synthesized images from generative models to augment training data has been studied before in many domains (Frid-Adar et al., 2018; Sankara-narayanan et al., 2018), including domain adaptation (Huang et al., 2018), visual alignment (Peebles et al., 2022), and mitigation of dataset bias (Sharmanska et al., 2020; Hemmat et al., 2023; Prabhu et al., 2024). For example, (Prabhu et al., 2024) introduces a methodology aimed at enhancing test set evaluation through augmentation. While previous methods predominantly relied on GANs (Zhang et al., 2021c; Li et al., 2022b; Tritrong et al., 2021) as the generative model, more recent studies promote using diffusion models to augment the data (Rombach et al., 2022; He et al., 2022b; Shipard et al., 2023; Trabucco et al., 2024; Azizi et al., 2023; Luo et al., 2023; Roy et al., 2023; Jain et al., 2022; Feng et al., 2023; Dunlap et al., 2023b; Chegini & Feizi, 2023). More specifically, (Trabucco et al., 2024; Roy et al., 2023; He et al., 2022b; Azizi et al., 2023) study the effectiveness of text-to-image diffusion models in data augmentation by diversification of each class with synthetic images. (Roy et al., 2023) also utilizes a text-to-image diffusion model, but with a BLIP (Li et al., 2022d) model to generate meaningful captions from the existing images. (Jain et al., 2022) utilizes diffusion models for augmentation to correct model mistakes. (Feng et al., 2023) uses CLIP (Radford et al., 2021) to filter generated images. Generative models for data augmentation may produce out-of-distribution samples if the downstream task's data distribution differs. Fine-tuning on a small downstream dataset can address this. For example, DAFusion (Trabucco et al., 2024) fine-tunes a diffusion model using textual inversion (Gal et al., 2022a), while SiSTA (Thopalli et al., 2023) adapts a GAN for the task. (Graikos et al., 2023a) propose adapting generative models to downstream tasks by leveraging the internal representations of the denoiser network. Investigations by (Tian et al., 2023) explore the use of text-to-image synthetic images for generating positive samples in contrastive learning. (Dunlap et al., 2023b) utilizes text-based diffusion and a large language model (LLM) to diversify the training data. (Chegini & Feizi, 2023) uses an LLM to generate text descriptions of failure modes associated with spurious correlations, which are then used to generate synthetic data through generative models. The challenge here is that the LLM has little understanding of such failure scenarios and contexts.

We take a completely different approach here, without replying on any extra source of information (e.g., through an LLM). Inspired by image editing approaches such as Boomerang (Luzi et al., 2022) and SDEdit (Meng et al., 2021), we propose to adaptively guide a latent diffusion model to generate *hard negatives* images (Mao et al., 2017; Xuan et al., 2021) on a per-sample basis per category. In a nutshell, the aforementioned studies focus on improving the diversity of each class with effective prompts and diffusion models, however, we focus on generating effective *hard negative* samples for each class by combining two sources of contradicting information (images from the source category and text prompt from the target category).

**Language Guided Recognition Models.** Vision-Language foundation models (VLMs) (Alayrac et al., 2022; Radford et al., 2021; Rombach et al., 2022; Saharia et al., 2022; Ramesh et al., 2022; 2021) utilize human language to guide the generation of images or to extract features from images that are aligned with human language. CLIP (Radford et al., 2021) excels in zero-shot tasks by aligning images with text, while recent works improve prompts (Dunlap et al., 2023a; Petryk et al., 2022) or use diffusion models as classifiers (Li et al., 2023). Similarly, we leverage Stable Diffusion 1.5 (Rombach et al., 2022) to enhance downstream tasks by augmenting training data with hard negative samples based on category names.

**Few-Shot Learning.** In Few-shot Learning (FSL), we pre-train a model with abundant data to learn a rich representation, then fine-tune it on new tasks with only a few available samples. In supervised FSL (Chen et al., 2019a; Afrasiyabi et al., 2019; Qiao et al., 2018; Ye et al., 2020; Dvornik et al., 2019; Li et al., 2020; Sung et al., 2018; Zhou et al., 2021; Singh & Jamali-Rad, 2023), pretraining is done on a labeled dataset, whereas in unsupervised FSL (Jang et al., 2022; Wang & Deng, 2022; Lu et al., 2022; Qin et al., 2020; Antoniou & Storkey, 2019; Khodadadeh et al., 2019; Hsu et al., 2018; Medina et al., 2020; Shirekar et al., 2023) the pretraining has to be conducted on an unlabeled dataset posing an extra challenge in the learning paradigm and neighboring these methods closer to the realm of self-supervised learning.

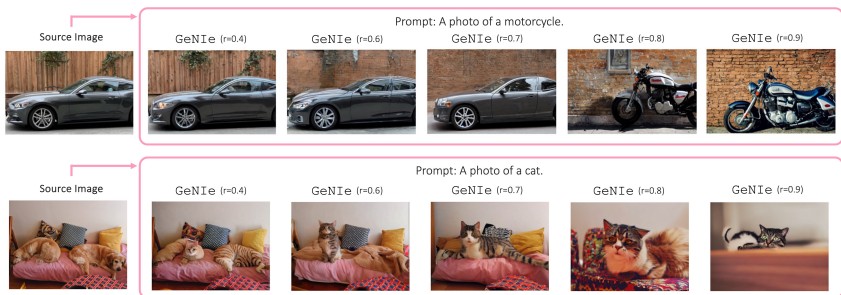

Figure 2: **Effect of noise ratio, $r$, in GeNIe:** we employ GeNIe to generate augmentations for the target classes (motorcycle and cat) with varying $r$. Smaller $r$ yields images closely resembling the source semantics, creating an inconsistency with the intended target label. By tracing $r$ from 0 to 1, augmentations gradually transition from source image characteristics to the target category. However, a distinct shift from the source to the target occurs at a specific $r$ that may vary for different source images or target categories. For more examples, please refer to Fig. A9.

## 3 PROPOSED METHOD: GeNIe

Given a source image $X_S$ from category S = <source category>, we are interested in generating a target image $X_r$ from category $T$ = <target category>. In doing so, we intend to ensure the low-level visual features or background context of the source image are preserved, so that we generate samples that would serve as *hard negatives* for the *source* image. To this aim, we adopt a conditional latent diffusion model (such as Stable Diffusion, (Rombach et al., 2022)) conditioned on a text prompt of the following format "A photo of a $T$ = <target category>".

**Key Idea.** GeNIe in its basic form is a simple yet effective augmentation sample generator for improving a classifier $f_\theta(.)$ with the following two key aspects: (i) inspired by (Luzi et al., 2022; Meng et al., 2021) instead of adding the full amount of noise $\sigma_{max}$ and going through all $N_{max}$ (being typically 50) steps of denoising, we use less amount of noise ($r\sigma_{max}$, with $r \in (0, 1)$) and consequently fewer number of denoising iterations ($\lfloor rN_{max} \rfloor$); (ii) we prompt the diffusion model with a $P$ mandating a target category $T$ different than the source $S$. Hence, we denote the conditional diffusion process as $X_r = \text{STDiff}(X_S, P, r)$. In such a construct, the proximity of the final decoded image $X_r$ to the source image $X_S$ or the target category defined through the text prompt $P$ depends on $r$. Hence, by controlling the amount of noise, we can generate images that blend characteristics of both the text prompt $P$ and the source image $X_S$. If we do not provide much of visual details in the text prompt (e.g., desired background, etc.), we expect the decoded image $X_r$ to follow the details of $X_S$ while reflecting the semantics of the text prompt $P$. We argue, and demonstrate later, that the newly generated samples can serve as *hard negative* examples for the source category $S$ since they share the low-level features of $X_S$ while representing the semantics of the target category, $T$. Notably, the source category $S$ can be randomly sampled or be carefully extracted from the confusion matrix of $f_\theta(.)$ based on real training data. The latter might result in even *harder negative* samples being now cognizant of model confusions. Finally, we will append our initial dataset with the newly generated hard negative samples through GeNIe and (re)train the classifier model.

**Enhancing GeNIe: GeNIe-Ada.** One of the remarkable aspects of GeNIe lies in its simple application, requiring only $X_S$, $P$, and $r$. However, selecting the appropriate value for $r$ poses a challenge as it profoundly influences the outcome. When $r$ is small, the resulting $X_r$ tends to closely resemble $X_S$, and conversely, when $r$ is large (closer to 1), it tends to resemble the semantics of the target category. This phenomenon arises because a smaller noise level restricts the capacity of the diffusion model to deviate from the semantics of the input $X_S$. Thus, a critical question emerges: how can we select $r$ for a particular source image to generate samples that preserve the low-level semantics of the source category $S$ in $X_S$ while effectively representing the semantics of the target category $T$? We propose a method to determine an ideal value for $r$.

Our intuition suggests that by varying the noise ratio $r$ from 0 to 1, $X_r$ will progressively resemble category $S$ in the beginning and category $T$ towards the end. However, somewhere between 0 and 1, $X_r$ will undergo a rapid transition from category $S$ to $T$. This phenomenon is empirically observed in our experiments with varying $r$, as depicted in Fig. 2. Although the exact reason for this rapid change remains uncertain, one possible explanation is that the intermediate points between two categories reside far from the natural image manifold, thus, challenging the diffusion model's

**Algorithm 1:** GeNIe-Ada

**Require:** $X_S, X_T, f_\theta(.), \text{STDiff}(.), M$
Extract $Z_S \leftarrow f_\theta(X_s), Z_T \leftarrow f_\theta(X_T)$
**for** $m \in [1, M]$ **do**
$\quad r \leftarrow \frac{m}{M}, Z_r \leftarrow f_\theta(\text{STDiff}(X, P, r))$
$\quad d_m \leftarrow \frac{(Z_r - Z_S)^T (Z_T - Z_S)}{||Z_T - Z_S||_2}$
$m^* \leftarrow \text{argmax}_m |d_m - d_{m-1}|, \forall m \in [2, M]$
$r^* \leftarrow \frac{m^*}{n}$
**Return:** $X_{r^*} = \text{STDiff}(X_S, P, r^*)$

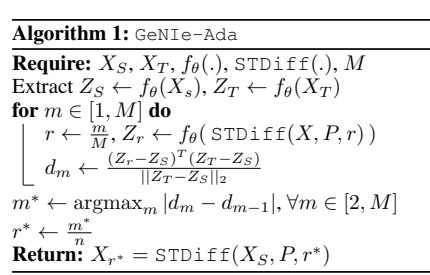 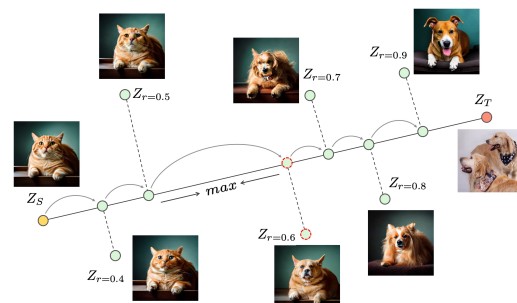

Figure 3: GeNIe-Ada: To choose $r$ adaptively for each (source image, target category) pair, we propose tracing the semantic trajectory from $Z_S$ (source image embeddings) to $Z_T$ (target embeddings) through backbone feature extractor $f_\theta(\cdot)$ (Algorithm 1). We adaptively select the sample right after the largest semantic shift.

capability to generate them. Ideally, we should select $r$ corresponding to just after this rapid semantic transition, as at this point, $X_r$ exhibits the highest similarity to the source image while belonging to the target category.

We propose to trace the semantic trajectory between $X_S$ and $X_T$ through backbone feature extractor $f_\theta(.)$. As shown in Algorithm 1, assuming access to the classifier backbone $f_\theta(.)$ and at least one example $X_T$ from the target category, we convert both $X_S$ and $X_T$ into their respective latent vectors $Z_S$ and $Z_T$ by passing them through $f_\theta(.)$. Then, we sample $M$ values for $r$ uniformly distributed $\in (0, 1)$, generating their corresponding $X_r$ and their latent vectors $Z_r$ for all those $r$. Subsequently, we calculate $d_r = \frac{(Z_r - Z_S)^T (Z_T - Z_S)}{||Z_T - Z_S||_2}$ as the distance between $Z_r$ and $Z_S$ projected onto the vector connecting $Z_S$ and $Z_T$. Our hypothesis posits that the rapid semantic transition corresponds to a sharp change in this projected distance. Therefore, we sample $n$ values for $r$ uniformly distributed between 0 and 1, and analyze the variations in $d_r$. We identify the largest gap in $d_r$ and select the $r$ value just after the gap when increasing $r$, as detailed in Algorithm 1 and illustrated in Fig. 3.

## 4 EXPERIMENTS

Since the impact of augmentation is more pronounced when the training data is limited, we evaluate the impact of GeNIe on Few-Shot classification in Section 4.1, Long-Tailed classification in Section 4.3, and fine-grained classification in Section 4.2. For GeNIe-Ada in all scenarios, we utilize GeNIe to generate augmentations from the noise level set $\{0.5, 0.6, 0.7, 0.8, 0.9\}$. The selection of the appropriate noise level per source image and target is adaptive, achieved through Algorithm 1.

**Baselines.** We use Stable Diffusion 1.5 (Rombach et al., 2022) as our base diffusion model. In all settings, we use the same prompt format to generate images for the target class: i.e., "A photo of a <target category>", where we replace the target category with the target category label. We generate $512 \times 512$ images for all methods. For fairness, we generate the same number of new images for each class. We use a single NVIDIA RTX 3090 for image generation. We consider 4 diffusion-based baselines and a suite of traditional data augmentation baselines.

**Img2Img** (Luzi et al., 2022; Meng et al., 2021): We sample an image from a target class, add noise to its latent representation and then pass it along with a prompt for the target category through reverse diffusion. The focus here is on a target class for which we generate extra positive samples. Adding large amount of noise leads to generating an image less similar to the original image. We use two different noise magnitudes for this baseline: $r = 0.3$ and $r = 0.7$ and denote them by $\text{Img2Img}^L$ and $\text{Img2Img}^H$, respectively.

**Txt2Img** (Azizi et al., 2023; He et al., 2022b): For this baseline, we omit the forward diffusion process and only use the reverse process starting from a text prompt for the target class of interest. This is similar to the base text-to-image generation strategy adopted in (Rombach et al., 2022; He et al., 2022b; Shipard et al., 2023; Azizi et al., 2023; Luo et al., 2023). Fig. 4 illustrates a set of generated augmentation examples for Txt2Img, Img2Img, and GeNIe.

DAFusion (Trabucco et al., 2024): In this method, an embedding is optimized with a set of images for each class to correspond to the classes in the dataset. This approach is introduced in Textual Inversion (Gal et al., 2022c). We optimize an embedding for 5000 iterations for each class in the dataset, followed by augmentation similar as the DAFusion method.

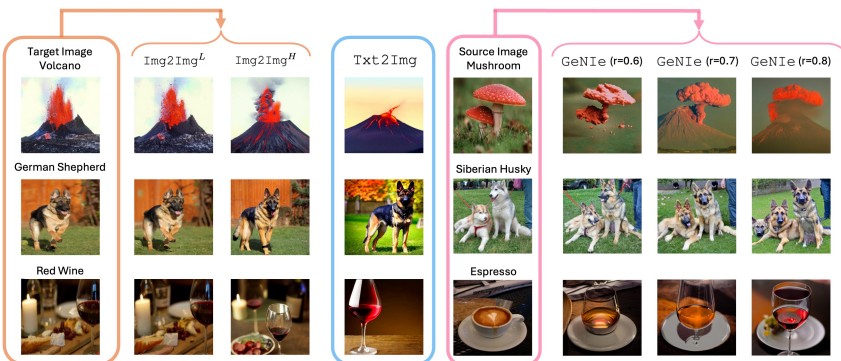

Figure 4: **Visualization of Generative Samples:** We compare `GeNIe` with two baselines: `Img2Img`$^L$ **augmentation:** both image and text prompt are from the same category. Adding noise does not change the image much, so they are not hard examples. `Txt2Img` **augmentation:** We simply use the text prompt only to generate an image for the desired category (e.g., using a text2image method). Such images may be far from the domain of our task since the generation is not informed by any visual data from our task. `GeNIe` **augmentation:** We use the target category name in the text prompt only along with the source image.

Cap2Aug(Roy et al., 2023): It is a recent diffusion-based data augmentation strategy that uses image captions as text prompts for an image-to-image diffusion model.

**Traditional Data Augmentation:** We consider both weak and strong traditional augmentations. More specifically, for weak augmentation we use random resize crop with scaling $\in [0.2, 1.0]$ and horizontal flipping. For strong augmentation, we consider random color jitter, random grayscale, and Gaussian blur. For the sake of completeness, we also compare against data augmentations such as CutMix (Yun et al., 2019) and MixUp (Zhang et al., 2018) that combine two images together.

## 4.1 FEW-SHOT CLASSIFICATION

We assess the impact of `GeNIe` compared to other augmentations in a number of few-shot classification (FSL) scenarios, where the model has to learn only from the samples contained in the ($N$-way, $K$-shot) support set and infer on the query set. Note that this corresponds to an inference-only FSL setting where a pretraining stage on an abundant dataset is discarded. The goal is to assess how well the model can benefit from the augmentations while keeping the original $N \times K$ samples intact.

**Datasets.** We conduct our few-shot experiments on two most commonly adopted few-shot classification datasets: *mini*-Imagenet (Ravi & Larochelle, 2017) and *tiered*-Imagenet (Ren et al., 2018). *mini*-Imagenet is a subset of ImageNet (Deng et al., 2009) for few-shot classification. It contains 100 classes with 600 samples each. We follow the predominantly adopted settings of (Ravi & Larochelle, 2017; Chen et al., 2019a) where we split the entire dataset into 64 classes for training, 16 for validation and 20 for testing. *tiered*-Imagenet is a larger subset of ImageNet with 608 classes and a total of 779, 165 images, which are grouped into 34 higher-level nodes in the *ImageNet* human-curated hierarchy. This set of nodes is partitioned into 20, 6, and 8 disjoint sets of training, validation, and testing nodes, and the corresponding classes form the respective meta-sets.

**Evaluation.** We evaluate the test-set accuracies of a state-of-the-art unsupervised few-shot learning method with `GeNIe` and compare them against the accuracies obtained using other augmentation methods. Specifically, we use UniSiam (Lu et al., 2022) pre-trained with ResNet-18, ResNet-34 and ResNet-50 backbones and follow its evaluation strategy of fine-tuning a logistic regressor to perform ($N$-way, $K$-shot) classification on the test sets of *mini*- and *tiered*-Imagenet. Following (Ravi & Larochelle, 2017), an episode consists of a labeled support-set and an unlabelled query-set. The support-set contains $N$ randomly sampled classes where each class contains $K$ samples, whereas the query-set contains $Q$ randomly sampled unlabeled images per class. We conduct our experiments on the two most commonly adopted settings: (5-way, 1-shot) and (5-way, 5-shot) classification settings. Following the literature, we sample 16-shots per class for the query set in both settings. We report the test accuracies along with the 95% confidence interval over 600 and 1000 episodes for *mini*-ImageNet and *tiered*-ImageNet, respectively.

**Implementation Details:** `GeNIe` generates augmented images for each class using images from all other classes as the source image. We use $r = 0.8$ in our experiments. We generate 4 samples per

Table 1: **mini-ImageNet:** We use our augmentations on (5-way, 1-shot) and (5-way, 5-shot) few-shot settings of mini-Imagenet dataset with 3 different backbones (ResNet-18, 34, and 50). We compare with various baselines and show that our augmentations with UniSiam outperform all the baselines including `Txt2Img` and DAFusion augmentation. The number of generated images per class is 4 for 1-shot and 20 for 5-shot settings.

**ResNet-18**

| Augmentation | Method | Pre-training | 1-shot | 5-shot |
|---|---|---|---|---|
| - | iDeMe-Net 2019b | sup. | 59.1±0.9 | 74.6±0.7 |
| - | Robust + dist 2019 | sup. | 63.7±0.6 | 81.2±0.4 |
| - | AFHN 2020 | sup. | 62.4±0.7 | 78.2±0.6 |
| Weak | ProtoNet+SSL 2020 | sup.+ssl | - | 76.6 |
| Weak | Neg-Cosine 2020 | sup. | 62.3±0.8 | 80.9±0.6 |
| - | Centroid Align 2019 | sup. | 59.9±0.7 | 80.4±0.7 |
| - | Baseline 2019a | sup. | 59.6±0.8 | 77.3±0.6 |
| - | Baseline++ 2019a | sup. | 59.0±0.8 | 76.7±0.6 |
| Weak | PSST 2021 | sup.+ssl | 59.5±0.5 | 77.4±0.5 |
| Weak | UMTRA 2019 | unsup. | 43.1±0.4 | 53.4±0.3 |
| Weak | ProtoCLR 2020 | unsup. | 50.9±0.4 | 71.6±0.3 |
| Weak | SimCLR 2020 | unsup. | 62.6±0.4 | 79.7±0.3 |
| Weak | SimSiam 2021 | unsup. | 62.8±0.4 | 79.9±0.3 |
| Weak | UniSiam+dist 2022 | unsup. | 64.1±0.4 | 82.3±0.3 |
| Weak | UniSiam 2022 | unsup. | 63.1±0.8 | 81.4±0.5 |
| Strong | UniSiam 2022 | unsup. | 62.8±0.8 | 81.2±0.6 |
| CutMix 2019 | UniSiam 2022 | unsup. | 62.7±0.8 | 80.6±0.6 |
| MixUp 2018 | UniSiam 2022 | unsup. | 62.1±0.8 | 80.7±0.6 |
| Img2Img$^L$ 2022 | UniSiam 2022 | unsup. | 63.9±0.8 | 82.1±0.6 |
| Img2Img$^H$ 2022 | UniSiam 2022 | unsup. | 69.1±0.7 | 84.0±0.5 |
| Txt2Img 2023; 2022b | UniSiam 2022 | unsup. | 74.1±0.6 | 84.6±0.5 |
| DAFusion 2024 | UniSiam 2022 | unsup. | 64.3±1.8 | 82.0±1.4 |
| GeNIe (Ours) | UniSiam 2022 | unsup. | 75.5±0.6 | 85.4±0.4 |
| GeNIe-Ada (Ours) | UniSiam 2022 | unsup. | 76.8±0.6 | 85.9±0.4 |

**ResNet-34**

| Augmentation | Method | Pre-training | 1-shot | 5-shot |
|---|---|---|---|---|
| Weak | Baseline 2019a | sup. | 49.8±0.7 | 73.5±0.7 |
| Weak | Baseline++ 2019a | sup. | 52.7±0.8 | 76.2±0.6 |
| Weak | SimCLR 2020 | unsup. | 64.0±0.4 | 79.8±0.3 |
| Weak | SimSiam 2021 | unsup. | 63.8±0.4 | 80.4±0.3 |
| Weak | UniSiam+dist 2022 | unsup. | 65.6±0.4 | 83.4±0.2 |
| Weak | UniSiam 2022 | unsup. | 64.3±0.8 | 82.3±0.5 |
| Strong | UniSiam 2022 | unsup. | 64.5±0.8 | 82.1±0.6 |
| CutMix 2019 | UniSiam 2022 | unsup. | 64.0±0.8 | 81.7±0.6 |
| MixUp 2018 | UniSiam 2022 | unsup. | 63.7±0.8 | 80.1±0.8 |
| Img2Img$^L$ 2022 | UniSiam 2022 | unsup. | 65.5±0.8 | 82.9±0.5 |
| Img2Img$^H$ 2022 | UniSiam 2022 | unsup. | 70.5±0.8 | 84.8±0.5 |
| Txt2Img 2023; 2022b | UniSiam 2022 | unsup. | 75.4±0.6 | 85.5±0.5 |
| DAFusion 2024 | UniSiam 2022 | unsup. | 64.7±1.9 | 83.2±1.4 |
| GeNIe (Ours) | UniSiam 2022 | unsup. | 77.1±0.6 | 86.3±0.4 |
| GeNIe-Ada (Ours) | UniSiam 2022 | unsup. | 78.5±0.6 | 86.6±0.4 |

**ResNet-50**

| Augmentation | Method | Pre-training | 1-shot | 5-shot |
|---|---|---|---|---|
| Weak | PDA+Net 2021 | unsup. | 63.8±0.9 | 83.1±0.6 |
| Weak | Meta-DM 2023 | unsup. | 66.7±0.4 | 85.3±0.2 |
| Weak | UniSiam 2022 | unsup. | 64.6±0.8 | 83.4±0.5 |
| Strong | UniSiam 2022 | unsup. | 64.8±0.8 | 83.2±0.5 |
| CutMix 2019 | UniSiam 2022 | unsup. | 64.3±0.8 | 83.2±0.5 |
| MixUp 2018 | UniSiam 2022 | unsup. | 63.8±0.8 | 84.6±0.5 |
| Img2Img$^L$ 2022 | UniSiam 2022 | unsup. | 66.0±0.8 | 84.0±0.5 |
| Img2Img$^H$ 2022 | UniSiam 2022 | unsup. | 71.1±0.7 | 85.7±0.5 |
| Txt2Img 2023; 2022b | UniSiam 2022 | unsup. | 76.4±0.6 | 86.5±0.4 |
| DAFusion 2024 | UniSiam 2022 | unsup. | 65.7±1.8 | 83.9±1.2 |
| GeNIe (Ours) | UniSiam 2022 | unsup. | 77.3±0.6 | 87.2±0.4 |
| GeNIe-Ada (Ours) | UniSiam 2022 | unsup. | 78.6±0.6 | 87.9±0.4 |

class as augmentations in the 5-way, 1-shot setting and 20 samples per class as augmentations in the 5-way, 5-shot setting. For the sake of a fair comparison, we ensure that the total number of labelled samples in the support set after augmentation remains the same across all different traditional and generative augmentation methodologies. Due to the expensive training of embeddings for each class in each episode, we only evaluated the DA-Fusion baseline on the first 100 episodes.

**Results:** The results on *mini*-Imagenet and *tiered*-Imagenet for both (5-way, 1 and 5-shot) settings are summarized in Table 1 and Table 3, respectively. Regardless of the choice of backbone, we observe that `GeNIe` helps consistently improve UniSiam's performance and outperform other supervised and unsupervised few-shot classification methods as well as other diffusion-based (Trabucco et al., 2024; Luzi et al., 2022; Rombach et al., 2021; He et al., 2022b) and classical (Yun et al., 2019; Zhang et al., 2018) data augmentation techniques on both datasets, across both (5-way, 1 and 5-shot) settings. Our noise adaptive method of selecting optimal augmentations per source image (`GeNIe-Ada`) further improves `GeNIe`'s performance across all three backbones, both few-shot settings, and both datasets (*mini* and *tiered*-Imagenet).

## 4.2 FINE-GRAINED FEW-SHOT CLASSIFICATION

To further investigate the impact of the proposed method, we compare `GeNIe` with other text-based data augmentation techniques across four distinct fine-grained datasets in a 20-way, 1-shot classification setting. We employ the pre-trained DINOV2 ViT-G (Oquab et al., 2023) backbone as a feature extractor to derive features from training images. Subsequently, an SVM classifier is trained on these features, and we report the Top-1 accuracy of the model on the test set.

**Results:** Table 2 summarizes the results. Additional details about this experiment can be found in Section A.8. `GeNIe` outperforms all other baselines, including `Txt2Img`, by margins upto 0.5% on CUB200, 6.6% on Cars196, 0.1% on Food101 and 5.3% on FGVC-Aircraft. `GeNIe` exhibits great effectiveness in more challenging datasets, outperforming the baseline with traditional augmentation by about 38% for the Cars dataset and by roughly 17% for the Aircraft dataset. It can be observed here that `GeNIe-Ada` performs on-par with `GeNIe` with a fixed noise level, eliminating the necessity for noise level search in `GeNIe`.

Table 2: **Few-shot Learning on Fine-grained dataset:** We utilize an SVM classifier trained atop the DINOV2 ViT-G pretrained backbone, reporting Top-1 accuracy for the test set of each dataset. The baseline is an SVM trained on the same backbone using weak augmentation.

| Method | Birds CUB200 2011 | Cars Cars196 2013 | Foods Food101 2014 | Aircraft Aircraft 2013 |
|---|---|---|---|---|
| Baseline | 90.3 | 49.8 | 82.9 | 29.2 |
| Img2Img$^L$ 2022 | 90.7 | 50.4 | 87.4 | 31.0 |
| Img2Img$^H$ 2022 | 91.3 | 56.4 | 91.7 | 34.7 |
| Txt2Img 2022b | 92.0 | 81.3 | 93.0 | 41.7 |
| GeNIe (r=0.5) | 92.0 | 84.6 | 91.5 | 39.8 |
| GeNIe (r=0.6) | 92.2 | 87.1 | 92.5 | 45.0 |
| GeNIe (r=0.7) | 92.5 | 87.9 | 92.9 | 47.0 |
| GeNIe (r=0.8) | 92.5 | 87.7 | 93.1 | 46.5 |
| GeNIe (r=0.9) | 92.4 | 87.1 | 93.1 | 45.7 |
| GeNIe-Ada | 92.6 | 87.9 | 93.1 | 46.9 |

Table 3: *tiered*-ImageNet: Accuracies (% ± std) for 5-way, 1-shot and 5-way, 5-shot classification settings on the test-set. We compare against various SOTA supervised and unsupervised few-shot classification baselines as well as other augmentation methods, with UniSiam 2022 pre-trained ResNet-18,50 backbones.

| ResNet-18 | | | | |
|---|---|---|---|---|
| Augmentation | Method | Pre-training | 1-shot | 5-shot |
| Weak | SimCLR 2020 | unsup. | 63.4±0.4 | 79.2±0.3 |
| Weak | SimSiam 2021 | unsup. | 64.1±0.4 | 81.4±0.3 |
| Weak | UniSiam 2022 | unsup. | 63.1±0.7 | 81.0±0.5 |
| Strong | UniSiam 2022 | unsup. | 62.8±0.7 | 80.9±0.5 |
| CutMix 2019 | UniSiam 2022 | unsup. | 62.1±0.7 | 78.9±0.6 |
| MixUp 2018 | UniSiam 2022 | unsup. | 62.1±0.7 | 78.4±0.6 |
| Img2Img$^L$ 2022 | UniSiam 2022 | unsup. | 63.9±0.7 | 81.8±0.5 |
| Img2Img$^H$ 2022 | UniSiam 2022 | unsup. | 68.7±0.7 | 83.5±0.5 |
| Txt2Img 2022b | UniSiam 2022 | unsup. | 72.9±0.6 | 84.2±0.5 |
| DAFusion 2024 | UniSiam 2022 | unsup. | 62.6±2.1 | 81.0±1.5 |
| GeNIe(Ours) | UniSiam 2022 | unsup. | **73.6±0.6** | **85.0±0.4** |
| GeNIe-Ada(Ours) | UniSiam 2022 | unsup. | **75.1±0.6** | **85.5±0.5** |
| ResNet-50 | | | | |
| Weak | PDA+Net 2021 | unsup. | 69.0±0.9 | 84.2±0.7 |
| Weak | Meta-DM 2023 | unsup. | 69.6±0.4 | 86.5±0.3 |
| Weak | UniSiam + dist 2022 | unsup. | 69.6±0.4 | 86.5±0.4 |
| Weak | UniSiam 2022 | unsup. | 66.8±0.7 | 84.7±0.5 |
| Strong | UniSiam 2022 | unsup. | 66.5±0.7 | 84.5±0.5 |
| CutMix 2019 | UniSiam 2022 | unsup. | 66.0±0.7 | 83.3±0.5 |
| MixUp 2018 | UniSiam 2022 | unsup. | 66.1±0.5 | 84.1±0.8 |
| Img2Img$^L$ 2022 | UniSiam 2022 | unsup. | 67.8±0.7 | 85.3±0.5 |
| Img2Img$^H$ 2022 | UniSiam 2022 | unsup. | 72.4±0.7 | 86.7±0.4 |
| Txt2Img 2022b | UniSiam 2022 | unsup. | 77.1±0.6 | 87.3±0.4 |
| DAFusion 2024 | UniSiam 2022 | unsup. | 66.5±2.2 | 84.8±1.4 |
| GeNIe (Ours) | UniSiam 2022 | unsup. | **78.0±0.6** | **88.0±0.4** |
| GeNIe-Ada (Ours) | UniSiam 2022 | unsup. | **78.8±0.6** | **88.6±0.6** |

Table 4: **Long-Tailed ImageNet-LT:** We compare different augmentation methods on ImageNet-LT and report Top-1 accuracy for "Few", "Medium", and "Many" sets. On the "Few" set and LiVT method, our augmentations improve the accuracy by 11.7 points compared to LiVT original augmentation and 4.4 points compared to Txt2Img. GeNIe-Ada outperforms Cap2Aug baseline in "Few" categories by 7.6%. Refer to Table A8 for a full comparison with prior Long-Tailed methods.

| ResNet-50 | | | | |
|---|---|---|---|---|
| Method | Many | Med. | Few | Overall Acc |
| ResLT 2022 | 63.3 | 53.3 | 40.3 | 55.1 |
| PaCo 2021b | 68.2 | 58.7 | 41.0 | 60.0 |
| LWS 2019 | 62.2 | 48.6 | 31.8 | 51.5 |
| Zero-shot CLIP 2021 | 60.8 | 59.3 | 58.6 | 59.8 |
| DRO-LT 2021 | 64.0 | 49.8 | 33.1 | 53.5 |
| VL-LTR 2022 | 77.8 | 67.0 | 50.8 | 70.1 |
| Cap2Aug 2023 | 78.5 | **67.7** | 51.9 | 70.9 |
| GeNIe-Ada | **79.2** | 64.6 | **59.5** | **71.5** |
| ViT-B | | | | |
| Method | Many | Med. | Few | Overall Acc |
| ViT 2021 | 50.5 | 23.5 | 6.9 | 31.6 |
| MAE 2022a | 74.7 | 48.2 | 19.4 | 54.5 |
| DeiT 2022 | 70.4 | 40.9 | 12.8 | 48.4 |
| LiVT 2023 | 73.6 | 56.4 | 41.0 | 60.9 |
| LiVT + Img2Img$^L$ | 74.3 | 56.4 | 34.3 | 60.5 |
| LiVT + Img2Img$^H$ | 73.8 | 56.4 | 45.3 | 61.6 |
| LiVT + Txt2Img | **74.9** | 55.6 | 48.3 | 62.2 |
| LiVT + GeNIe-Ada | 74.0 | **56.9** | **52.7** | **63.1** |

## 4.3 LONG-TAILED CLASSIFICATION

We evaluate our method on long-tailed data, where the number of instances per class is unbalanced, with most categories having limited samples (tail). Our goal is to mitigate this bias by augmenting the tail of the distribution with generated samples. We evaluate GeNIe using two backbones: ViT with LViT (Xu et al., 2023) and ResNet50 with VL-LTR (Tian et al., 2022). Following LViT, we first train an MAE (He et al., 2021) and ViT on the unbalanced dataset without any augmentation. Next, we train the Balanced Fine-Tuning stage of LViT by incorporating the augmentation data generated using GeNIe or other baselines. For ResNet50, we use VL-LTR code to fine-tune the CLIP ResNet50 with generated augmentations by GeNIe.

**Dataset:** We perform experiments on ImageNet-LT (Liu et al., 2019). It contains 115.8K images from $1,000$ categories. The number of images per class varies from 1280 to 5. Imagenet-LT classes can be divided into 3 groups: "Few" with less than 20 images, "Med" with $20 - 100$ images, and "Many" with more than 100 images. Imagenet-LT uses the same validation set as ImageNet. We augment "Few" categories only and limit the number of generated images to 50 samples per class. For GeNIe, instead of randomly sampling the source images from other classes, we use a confusion matrix on the training data to find the top-4 most confused classes and only consider those classes for random sampling of the source image.

**Results:** Augmenting training data with GeNIe-Ada improves accuracy on the "Few" set by 11.7% and 4.4% compared with LViT only and LViT with Txt2Img augmentation baselines respectively. In ResNet50, GeNIe-Ada outperforms Cap2Aug baseline in "Few" categories by 7.6%. The results are summarized in Table 4. Please refer to Section A.10 for implementation details.

## 4.4 ABLATION AND FURTHER ANALYSIS

**Semantic Shift from Source to Target Class.** The core motivation behind GeNIe-Ada is that by varying the noise ratio $r$ from 0 to 1, augmented sample $X_r$ will progressively shift its semantic category from source ($S$) in the beginning to target category ($T$) towards the end. However, somewhere between 0 and 1, $X_r$ will undergo a rapid transition from $S$ to $T$. To demonstrate this hypothesis empirically, in Figs. 5 and A7, we visualize pairs of source images and target categories with their respective GeNIe generated augmentations for different noise ratios $r$, along with their corresponding PCA-projected embedding scatter plots (on the far left). We extract embeddings for all the images using a DINOv2 ViT-G pretrained backbone, which we assume as an oracle model in identifying the right category. We observe that as $r$ increases from 0.3 to 0.8, the images transition to embody

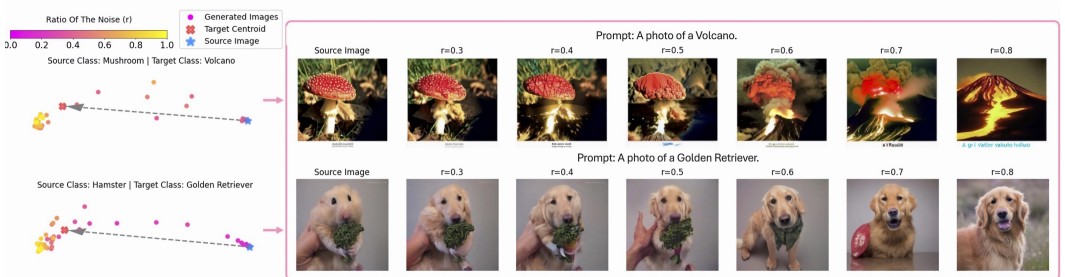

Figure 5: **Embedding visualizations of generative augmentations:** We pass all generative augmentations through DINOv2 ViT-G (serving as an oracle) to extract their corresponding embeddings and visualize them with PCA. As shown, the extent of semantic shifts varies based on both the source image and the target class.

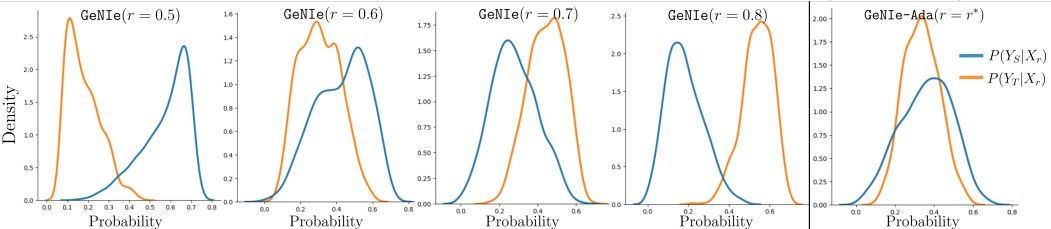

Figure 6: **Why `GeNIe` augmentations are challenging?** While deciding which class the generated augmentations ($X_r$) belong to is already difficult within $r = [0.6, 0.7]$ (due to high overlap between $P(Y_S|X_r)$ and $P(Y_T|X_r)$), `GeNIe-Ada` selects the best noise threshold ($r^*$) offering the hardest negative sample.

more of the target category's semantics while preserving the contextual features of the source image. This transition of semantics can also be observed in the embedding plots (on the left) where they consistently shift from the proximity of the source image (blue star) to the target class's centroid (red cross) as the noise ratio $r$ increases. The sparse distribution of points within $r = [0.4, 0.6]$ for the first image and $r = [0.2, 0.4]$ for the second image aligns with our intuition of a rapid transition from category $S$ to $T$, thus empirically affirming our motivation behind `GeNIe-Ada`.

To further establish this, in Fig. 6, we demonstrate the efficacy of `GeNIe` in generating hard negatives at the decision boundaries of an SVM classifier, which is trained on the labelled support set of the few-shot tasks of *mini*-Imagenet, without any augmentations. We then plot source and target class probabilities ($P(Y_S|X_r)$ and $P(Y_T|X_r)$, respectively) of the generated augmentation samples $X_r$. For both $r = 0.6$ and $0.7$, there is significant overlap between $P(Y_S|X_r)$ and $P(Y_T|X_r)$, making it difficult for the classifier to decide the correct class. On the right-hand-side, `GeNIe-Ada` automatically selects the best $r$ resulting in the most overlap between the two distributions, thus offering the hardest negative sample among the considered $r$ values (for more details see A.1). Note that a large overlap between distributions is not sufficient to call the generated samples hard negatives because they should also belong to the target category. This is, however, confirmed by the high Oracle accuracy in Table 5 (elaborated in detail in the following paragraph) which verifies that majority of the generated augmentation samples do belong to the target category.

**Label consistency of the generated samples.** The choice of noise ratio $r$ is important in producing hard negative examples. In Table 5, we present the accuracy of the `GeNIe` model across various noise ratios, alongside the oracle accuracy, which is an ImageNet pre-trained DeiT-Base (Touvron et al., 2021b) classifier. We observe a decline in the label consistency of generated data (quantified by the performance of the oracle model) when decreasing the noise level. Reducing $r$ also results in a degradation in the performance of the final few-shot model ($87.2\% \rightarrow 77.6\%$) corroborating that an appropriate choice of $r$ plays a crucial role. We investigate this further in the following paragraph.

**Effect of Noise in `GeNIe`.** We examine the impact of noise on the performance of the few-shot model in Table 5. Noise levels $r \in [0.7, 0.8]$ yield the best performance. Conversely, utilizing noise levels below $0.7$ diminishes performance due to label inconsistency, as is demonstrated in Table 5 and Fig 5. As such, determining the appropriate noise level is pivotal for the performance of `GeNIe` to be able to generate challenging hard negatives while maintaining label consistency. An alternative approach to finding the optimal noise level involves using `GeNIe-Ada` to adaptively select the noise level for each source image and target class. As demonstrated in Tables 5 and 2, `GeNIe-Ada` matches or outperforms `GeNIe` with fixed noise levels.

Table 5: **Effect of Noise and Diffusion Models in `GeNIe`:** We use the same setting as in Table 1 to study the effect of the amount of noise. As expected (also shown in Fig 5), small noise results in worse accuracy since some generated images may be from the source category rather than the target one. For $r = 0.5$ only 73% of the generated data is from the target category. This behaviour is also shown in Fig. 2. Notably, reducing the noise level below 0.7 is associated with a decline in oracle accuracy and subsequent degradation in the performance of the final few-shot model. Note that the high oracle accuracy of `GeNIe-Ada` demonstrates its capability to adaptively select the noise level per source and target, ensuring semantic consistency with the intended target. To further demonstrate `GeNIe`'s ability to generalize across different diffusion models, we replace the diffusion model with SD3 and SDXL-Turbo. The resulting accuracies follow a similar trend to those in Table 1, confirming `GeNIe`'s advantage over `Txt2Img` across various diffusion models.

| Method | Generative Model | Noise r= | ResNet-18 | | ResNet-34 | | ResNet-50 | | Oracle Acc |
|---|---|---|---|---|---|---|---|---|---|
| | | | 1-shot | 5-shot | 1-shot | 5-shot | 1-shot | 5-shot | |
| Txt2Img | SD 1.5 | - | 74.1±0.6 | 84.6±0.5 | 75.4±0.6 | 85.5±0.5 | 76.4±0.6 | 86.5±0.4 | - |
| GeNIe | SD 1.5 | 0.5 | 60.4±0.8 | 74.1±0.6 | 62.0±0.8 | 75.8±0.6 | 63.7±0.9 | 77.6±0.6 | 73.4±0.5 |
| GeNIe | SD 1.5 | 0.6 | 69.7±0.7 | 80.7±0.5 | 71.1±0.7 | 82.2±0.5 | 72.1±0.7 | 82.8±0.5 | 85.8±0.4 |
| GeNIe | SD 1.5 | 0.7 | 74.5±0.6 | 83.3±0.5 | 76.4±0.6 | 84.4±0.5 | 77.1±0.6 | 85.0±0.4 | 94.5±0.2 |
| GeNIe | SD 1.5 | 0.8 | 75.5±0.6 | 85.4±0.4 | 77.1±0.6 | 86.3±0.4 | 77.3±0.6 | 87.2±0.4 | 98.2±0.1 |
| GeNIe | SD 1.5 | 0.9 | 75.0±0.6 | 85.3±0.4 | 77.6±0.6 | 86.2±0.4 | 77.7±0.6 | 87.0±0.4 | 99.3±0.1 |
| GeNIe-Ada | SD 1.5 | Adaptive | 76.8±0.6 | 85.9±0.4 | 78.5±0.6 | 86.6±0.4 | 78.6±0.6 | 87.9±0.4 | 98.9±0.2 |
| Txt2Img | SDXL-Turbo | - | 72.5±0.3 | 82.1±0.6 | 76.2±0.2 | 84.4±0.3 | 76.7±0.6 | 85.9±0.5 | - |
| GeNIe | SDXL-Turbo | 0.5 | 61.2±0.5 | 73.5±0.2 | 61.5±0.2 | 74.9±0.3 | 63.1±0.2 | 76.5±0.6 | - |
| GeNIe | SDXL-Turbo | 0.6 | 70.2±0.2 | 79.3±0.4 | 71.2±0.7 | 81.4±0.6 | 73.2±0.2 | 82.4±0.5 | - |
| GeNIe | SDXL-Turbo | 0.7 | 73.1±0.3 | 83.5±0.5 | 76.1±0.6 | 85.3±0.4 | 77.2±0.6 | 84.2±0.4 | - |
| GeNIe | SDXL-Turbo | 0.8 | 74.2±0.3 | 85.1±0.3 | 76.9±0.4 | 85.5±0.5 | 78.7±0.6 | 87.7±0.4 | - |
| GeNIe | SDXL-Turbo | 0.9 | 73.9±0.4 | 84.9±0.7 | 76.6±0.7 | 84.2±0.6 | 78.1±0.5 | 87.0±0.4 | - |
| GeNIe-Ada | SDXL-Turbo | Adaptive | 75.1±0.3 | 87.1±0.8 | 78.9±0.5 | 85.2±0.5 | 79.0±0.6 | 88.6±0.2 | - |
| Txt2Img | SD 3 | - | 73.6±1.7 | 82.9±1.2 | 76.7±1.5 | 85.5±1.3 | 77.2±1.9 | 85.0±1.2 | - |
| GeNIe | SD 3 | 0.5 | 62.0±1.2 | 72.9±1.1 | 62.5±0.9 | 73.9±1.0 | 64.1±0.5 | 76.1±1.9 | - |
| GeNIe | SD 3 | 0.6 | 70.8±1.5 | 79.1±1.9 | 71.8±1.2 | 82.1±1.3 | 74.1±1.5 | 83.4±1.8 | - |
| GeNIe | SD 3 | 0.7 | 74.6±0.8 | 84.5±1.2 | 76.5±1.9 | 86.2±1.6 | 78.5±1.9 | 84.0±1.1 | - |
| GeNIe | SD 3 | 0.8 | 75.9±1.2 | 86.3±1.7 | 77.8±1.9 | 85.5±1.9 | 79.2±1.7 | 88.3±1.9 | - |
| GeNIe | SD 3 | 0.9 | 75.1±0.5 | 85.2±1.2 | 78.1±1.3 | 86.2±1.2 | 77.1±1.9 | 88.9±0.8 | - |
| GeNIe-Ada | SD 3 | Adaptive | 76.8±1.3 | 87.5±1.5 | 78.9±1.3 | 87.7±1.5 | 79.1±1.4 | 89.5±1.0 | - |

**Effect of Diffusion Models in `GeNIe`.** We have tried experimenting with both smaller as well as more recent diffusion models. More specifically, we have used Stable Diffusion XL-Turbo to generate hard-negatives through `GeNIe` and `GeNIe-Ada`. Few-shot classification results on mini-Imagenet with these augmentations are shown in Table 5. The accuracies follow a similar trend to that of Table 1, where Stable Diffusion 1.5 was used to generate augmentations. `GeNIe-Ada` improves UniSiam's few-shot performance the most as compared to `GeNIe` with different noise ratios $r$, and even when compared to `Txt2Img`. This empirically indicates the robustness of `GeNIe` and `GeNIe-Ada` to different diffusion engines. Note that, Stable Diffusion XL-Turbo by default uses 4 steps for the sake of optimization, and to ensure we can have the right granularity for the choice of $r$ we have set the number of steps to 10. That is already 5 times faster than the standard Stable Diffusion v1.5 with 50 steps. Our experiments with Stable Diffusion v3 (which is a totally different model with a Transformers backbone) also in Table 5 also convey the same message. As such, we believe our approach is generalizable across different diffusion models.

## 5 CONCLUDING REMARKS

`GeNIe`, for the first time to our knowledge, combines contradictory sources of information (a source image, and a different target category prompt) through a noise adjustment strategy into a conditional latent diffusion model to generate challenging augmentations, which can serve as hard negatives.

**Limitation.** The required time to create augmentations through `GeNIe` is on par with any typical diffusion-based competitors (Azizi et al., 2023; He et al., 2022b); however, this is naturally slower than traditional augmentation techniques (Yun et al., 2019; Zhang et al., 2018). This is not a bottleneck in offline augmentation strategies, but can be considered a limiting factor in real-time scenarios. Recent studies are already mitigating this through advancements in diffusion model efficiency (Sauer et al., 2023; Meng et al., 2023; Liu et al., 2023). Another challenge present in any generative AI-based augmentation technique is the domain shift between the distribution of training data and the downstream context they might be used for augmentation. A possible remedy is to fine-tune the diffusion backbone on a rather small dataset from the downstream task.

**Broader Impact.** `GeNIe` can have a significant impact when it comes to generating challenging augmentations and thus enhancing downstream tasks beyond classification. Like any other generative model, `GeNIe` can also introduce inherent biases stemming from the training data used to build its diffusion backbone, which can reflect and amplify societal prejudices or inaccuracies. Therefore, it is crucial to carefully mitigate potential biases in generative models such as `GeNIe` to ensure a fair and ethical deployment of deep learning systems.

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

# A APPENDIX

## A.1 ANALYZING GeNIe, GeNIe-Ada'S CLASS-PROBABILITIES

The core aim of `GeNIe` and `GeNIe-Ada` is to address the failure modes of a classifier by generating *challenging* samples located near the decision boundary of each class pair, which facilitates the learning process in effectively enhancing the decision boundary between classes. As summarized in Table 5 and illustrated in Fig. 5, we have empirically corroborated that `GeNIe` and `GeNIe-Ada` can respectively produce samples $X_r, X_{r^*}$ that are negative with respect to the source image $X_S$, while semantically belonging to the class $T$. To

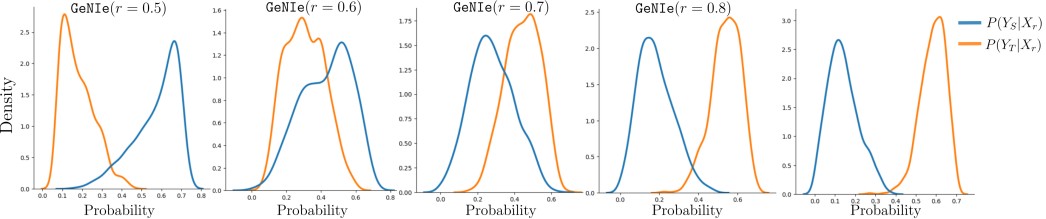

Figure A1: $P(Y_S|X_r)$ and $P(Y_T|X_r)$ for $r \in \{0.5, 0.6, 0.7, 0.8, 0.9\}$. On average, the classifier confidently predicts the source class more than the target class for $X_r$ for $r = 0.5$, and vice-versa for $r = 0.8, 0.9$. However, for $r = 0.6, 0.7$, the classifier struggles to classify $X_r$, indicating that the augmented samples are located closer to the decision boundary.

further analyze the effectiveness of `GeNIe` and `GeNIe-Ada`, we compare the source class-probabilities $P(Y_S|X_r)$ and target-class probabilities $P(Y_S|X_r)$ of augmented samples $X_r$.

To compute these class probabilities, we first fit an SVM classifier (as followed in UniSiam (Lu et al., 2022)) only on the labelled support set embeddings of each episode in the *mini*Imagenet test dataset. Then, we perform inference using each episode's SVM classifier on its respective $X_r$'s and extract its class probabilities of belonging to its source class $S$ and target class $T$. These per augmentation-sample source and target class probabilities are then averaged for each episode for each $r \in \{0.5, 0.6, 0.7, 0.8, 0.9\}$ in the case of `GeNIe` and for the optimal $r = r^*$ per sample in the case of `GeNIe-Ada`, plotted as density plots in Fig. A1, Fig. A2, respectively. Fig. A1 illustrates that $P(Y_S|X_r)$ and $P(Y_T|X_r)$ have significant overlap in the case of $r \in \{0.6, 0.7\}$ indicating class-confusion for $X_r$.

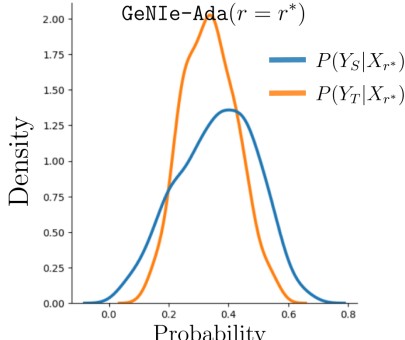

Furthermore, Fig. A2 illustrates that when using the optimal $r = r^*$ found by `GeNIe-Ada` per sample, $P(Y_S|X_r)$ and $P(Y_T|X_r)$ significantly overlap around probability scores of $0.2-0.45$, indicating class confusion for `GeNIe-Ada` augmentations. This corroborates with our analysis in Section 4.4, Table 5 and additionally empirically proves that the augmented samples generated by `GeNIe` for $r \in \{0.6, 0.7\}$ and `GeNIe-Ada` for $r = r^*$ are actually located near the decision boundary of each class pair.

Figure A2: Significant overlap between $P(Y_S|X_{r^*})$ and $P(Y_T|X_{r^*})$ indicates high class-confusion for augmented samples generated by `GeNIe-Ada`.

## A.2 INDEPENDENCE OF GENERATED AUGMENTATIONS FROM DOWNSTREAM TEST SETS

Here we analyzed whether the augmented samples generated by `GeNIe` using the diffusion model overlap with the test set of the downstream task. To set the stage, we extracted the latent embeddings corresponding to the train set (i.e., support), test set (i.e., query), and augmentations generated by `GeNIe`. Fig A3 illustrates the distribution of distances between train-test and augmentation-test pairs across 600 episodes. Notably, the mean distance of augmentation-test pairs is higher than that of train-test pairs, indicating that the augmented samples are distinct from the test set. This observation aligns with the fundamental assumption of train and test sets being mutually exclusive. Additionally, Fig A3 provides further evidence through a UMAP embedding plot of a randomly selected episode, where the embeddings of train, test, and augmented samples are visualized. The

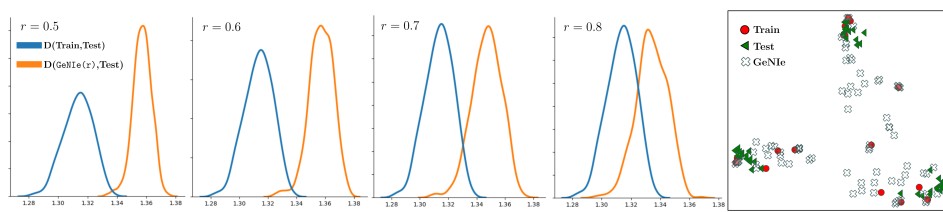

Figure A3: Comparison of embedding distributions and UMAP visualization for train, test, and GeNIe-augmented samples.

plot reveals clear separations between the test set and augmented samples, further confirming that the augmented samples do not overlap with or resemble the test set in embedding space. These findings validate that the diffusion-generated augmentations are independent of the downstream task's test set, ensuring the integrity of the evaluation process.

## A.3 ADDITIONAL AUGMENTATION COMPARISONS

We compute few-shot classification scores on mini-ImageNet with additional combinations of traditional augmentations. We introduce a Mixed augmentation scheme where we use a combination of Weak and Strong augmentations together. We also experiment the scenario where CutMix and MixUp are used alongside the Mixed augmentation strategy as indicated by Mixed+CutMix and Mixed+MixUp. Finally, we experiment with a combination of GeNIe along with MixUp, similar to (Graikos et al., 2023b). As can be seen in Tab. A1, we notice marginal improvements of upto 0.6% by using the Mixed augmentations either with or without the CutMix, MixUp counterparts. We also notice a drop in performance of upto 0.9% when MixUp is used along with GeNIe. This follows the general trend of drop in performance when using CutMix or MixUp, as reported in Tab. 1.

Table A1: *mini*-**ImageNet:** We use our augmentations on (5-way, 1-shot) and (5-way, 5-shot) few-shot settings of mini-Imagenet dataset with 2 different backbones (ResNet-18 and 50). We compare with additional combinations of traditional augmentations, with and without GeNIe. The number of generated images per class is 4 for 1-shot and 20 for 5-shot settings.

| ResNet-18 | | | | |
|---|---|---|---|---|
| **Augmentation** | **Method** | **Pre-training** | **1-shot** | **5-shot** |
| Weak | UniSiam 2022 | unsup. | 63.1±0.8 | 81.4±0.5 |
| Strong | UniSiam 2022 | unsup. | 62.8±0.8 | 81.2±0.6 |
| Mixed | UniSiam 2022 | unsup. | 63.2±0.5 | 81.9±0.4 |
| CutMix 2019 | UniSiam 2022 | unsup. | 62.7±0.8 | 80.6±0.6 |
| MixUp 2018 | UniSiam 2022 | unsup. | 62.1±0.8 | 80.7±0.6 |
| Mixed+MixUp 2018 | UniSiam 2022 | unsup. | 65.7±0.9 | 82.1±0.2 |
| Mixed+CutMix 2018 | UniSiam 2022 | unsup. | 64.9±0.8 | 81.6±0.5 |
| DAFusion 2024 | UniSiam 2022 | unsup. | 64.3±1.8 | 82.0±1.4 |
| GeNIe+MixUp | UniSiam 2022 | unsup. | 74.8±0.5 | 84.5±0.3 |
| GeNIe (Ours) | UniSiam 2022 | unsup. | 75.5±0.6 | 85.4±0.4 |
| GeNIe-Ada (Ours) | UniSiam 2022 | unsup. | **76.8±0.6** | **85.9±0.4** |

| ResNet-50 | | | | |
|---|---|---|---|---|
| **Augmentation** | **Method** | **Pre-training** | **1-shot** | **5-shot** |
| Weak | UniSiam 2022 | unsup. | 64.6±0.8 | 83.4±0.5 |
| Strong | UniSiam 2022 | unsup. | 64.8±0.8 | 83.2±0.5 |
| Mixed | UniSiam 2022 | unsup. | 64.5±0.5 | 83.8±0.5 |
| CutMix 2019 | UniSiam 2022 | unsup. | 64.3±0.8 | 83.2±0.5 |
| MixUp 2018 | UniSiam 2022 | unsup. | 63.8±0.8 | 84.6±0.5 |
| Mixed+MixUp 2018 | UniSiam 2022 | unsup. | 64.9±0.7 | 84.5±0.7 |
| Mixed+CutMix 2018 | UniSiam 2022 | unsup. | 63.5±0.5 | 83.0±0.8 |
| DAFusion 2024 | UniSiam 2022 | unsup. | 65.7±1.8 | 83.9±1.2 |
| GeNIe+MixUp | UniSiam 2022 | unsup. | 76.4±0.5 | 85.9±0.7 |
| GeNIe (Ours) | UniSiam 2022 | unsup. | 77.3±0.6 | 87.2±0.4 |
| GeNIe-Ada (Ours) | UniSiam 2022 | unsup. | **78.6±0.6** | **87.9±0.4** |

## A.4 EFFECT OF BACKBONE FOR NOISE RATIO SELECTOR IN GeNIe-Ada

To analyze the effect of the backbone feature extractor $f_\theta$ on selecting the optimal hard-negative using GeNIe-Ada, we use a pre-trained DeiT-B (Touvron et al., 2021a) instead of the UniSiam pretrained ResNet backbone. However, we still utilize the same ResNet backbone for few-shot classification. As shown in Tab. A2, we notice a marginal improvement of upto 0.7% when using GeNIe-Ada+DeiT-B as compared to GeNIe-Ada which uses the UniSiam pre-trained ResNet backbone. This suggests that there is still potential to develop more effective strategies for selecting noise ratios to further enhance GeNIe. However, in this paper, we limit our exploration to GeNIe-Ada and leave these improvements for future work.

## A.5 PSUEDOCODE OF GeNIe:

As illustrated in Alg. 2, we provide a detailed pytorch-style pseudocode for GeNIe. First, a SDv1.5 pipeline initialized by loading all the components such as the VQ-VAE encoder and decoder, the CLIP text encoder and the DPM scheduler for the forward and reverse diffusion process. Then, the source image is input to the encoder to encode the image into latent space for the diffusion model.

Table A2: **Effect of Backbone for Noise Ratio Selector in `GeNIe-Ada`:** We evaluate the impact of the noise ratio selector used in `GeNIe-Ada` ($f_\theta(.)$). Note that in all experiments presented in this paper, we use the same backbone for $f_\theta(.)$ that is subsequently fine-tuned for few-shot classification tasks. However, to analyze the effect of $f_\theta(.)$ on sampled augmentations, we replace it with a more powerful backbone, specifically DeiT-B pretrained on ImageNet-1K. It is important to note that this is not a practical assumption; if DeiT-B were available for noise selection, it could also be used as the classifier in few-shot experiments, outperforming the weaker backbones employed in our study. Nevertheless, this experiment demonstrates that using a stronger backbone can result in more accurate selection of augmentations in `GeNIe`, thereby enhancing the final accuracy. To clarify, DeiT-B is utilized solely as $f_\theta(.)$ for sampling augmentations and not as the classifier. Therefore, the observed improvement is attributed exclusively to better augmentation sampling.

| ResNet-18 | | | | | ResNet-50 | | | | |
|---|---|---|---|---|---|---|---|---|---|
| **Augmentation** | **Noise Ratio Selector Backbone $f_\theta(.)$** | **Method [Classifier Backbone]** | **1-shot** | **5-shot** | **Augmentation** | **Noise Selector Backbone $f_\theta(.)$** | **Method [Classifier Backbone]** | **1-shot** | **5-shot** |
| GeNIe (Ours) | - | UniSiam[ResNet18] | 75.5±0.6 | 85.4±0.4 | GeNIe | - | UniSiam[ResNet50] | 77.3±0.6 | 87.2±0.4 |
| GeNIe-Ada | UniSiam[ResNet18] | UniSiam[ResNet18] | 76.8±0.6 | 85.9±0.4 | GeNIe-Ada | UniSiam[ResNet50] | UniSiam[ResNet50] | 78.6±0.6 | 87.9±0.4 |
| GeNIe-Ada | IN-1K[DeiT-B] | UniSiam[ResNet18] | 77.5±0.5 | 86.3±0.2 | GeNIe-Ada | IN-1K[DeiT-B] | UniSiam[ResNet50] | 79.2±0.4 | 88.3±0.5 |

Next, the encoded image is partially noised based on the noise ratio $r$ using the scheduler. The diffusion model then de-noises the partially noised latent embedding for a total of NUM INFERENCE STEPS $\times r$ steps, with an additional input of a text prompt from a contradictory target class. Finally, the decoder decodes the de-noised latent embedding into the generated hard-negative image, that contains the low-level features of the source image and the class/category of the contradictory text-prompt.

---

**Algorithm 2:** PyTorch-style Pseudocode of GeNIe.

```python
# StableDiffusionPipeline: Pre-trained diffusion model
# DPMSolverMultistepScheduler: Scheduler for forward and reverse diffusion
# encode_latents: Encodes an image into latent space
# decode_latents: Decodes latents back into an image

def AugmentGeNIe(source_image, target_prompt, percent_noise):
    NUM_INFERENCE_STEPS = 50 # Number of steps for reverse diffusion
    NUM_TRAIN_STEPS = 1000 # Number of steps for forward diffusion

    # Initialize the stable diffusion pipeline and scheduler
    pipe = StableDiffusionPipeline.from_pretrained("stable-diffusion-v1-5")
    scheduler = DPMSolverMultistepScheduler.from_config(pipe.scheduler.config)

    # Encode the source image into latent space
    latents = encode_latents(source_image)

    # Forward Diffusion
    noise = torch.randn(latents.shape) # Generate random noise
    timestep = torch.Tensor([int(NUM_TRAIN_STEPS * percent_noise)]) # Calculate timestep
    latents_noise = scheduler.add_noise(latents, noise, timestep) # Add noise to latents

    # Reverse Diffusion
    latents = pipe(
        prompt=target_prompt,
        percent_noise=percent_noise,
        latents=latents_noise,
        num_inference_steps=NUM_INFERENCE_STEPS
    )

    # Decode latents back into an augmented image
    augmented_image = decode_latents(latents)

    return augmented_image
```

---

## A.6    IMPACT OF GENIE WITH FINE-TUNING:

For all our experiments regarding `GeNIe` and `GeNIe-Ada`, we assume that the base diffusion model is aware/has been trained on some samples of the target class. This facilitates the addition of the target class (input as text-prompt) into the generated augmentation, while retaining the low-level features of the source image through partial noising. However, there can be a scenario where the base diffusion model does not understand the contradictory text prompt and thus fails to incorporate it into the generated image. As a solution, we can use textual inversion (Gal et al., 2022b) to fine-tune the diffusion model on few images belonging the unknown target class to learn the corresponding embeddings for the target categories. This fine-tuning allows us to learn embeddings specific to the target class, enabling the generation of the desired hard-negative examples. To

empirically demonstrate the robustness of `GeNIe` on these scenarios, we present few-shot classification results on mini-Imagenet using `GeNIe` hard-negative augmentations in Tab. A3, generated by textual-inversion fine-tuning the diffusion model on images of the target class. Note that once the diffusion model is fine-tuned, the procedure to generate hard-negatives using partial noising and a contradictory text-prompt remains the same. As can be seen in Tab. A3, `GeNIe`+TxtInv performs significantly better than DAFusion baseline. It is important to note that, in this case, we do not utilize any information about the target category labels. DAFusion also employs textual-inversion-based fine-tuning; however, it does so without generating hard-negative samples. This indicates that GeNIe is effective even in scenario where the diffusion model is unaware of the target-class.

Table A3: ***mini*-ImageNet:** We use our augmentations on (5-way, 1-shot) and (5-way, 5-shot) few-shot settings of mini-Imagenet dataset with 2 different backbones (ResNet-18 and 50), by using Textual-Inversion (Gal et al., 2022b) on the target-classes. The number of generated images per class is 4 for 1-shot and 20 for 5-shot settings.

| ResNet-18 | | | | | ResNet-50 | | | | |
|---|---|---|---|---|---|---|---|---|---|
| **Augmentation** | **Method** | **Pre-training** | **1-shot** | **5-shot** | **Augmentation** | **Method** | **Pre-training** | **1-shot** | **5-shot** |
| DAFusion 2024 | UniSiam 2022 | unsup. | 64.3±1.8 | 82.0±1.4 | DAFusion 2024 | UniSiam 2022 | unsup. | 65.7±1.8 | 83.9±1.2 |
| GeNIe+TxtInv | UniSiam 2022 | unsup. | **73.9±0.8** | **84.6±0.9** | GeNIe+TxtInv | UniSiam 2022 | unsup. | **76.2±1.2** | **86.2±0.9** |

### A.7 COMPUTATIONAL COMPLEXITY OF `GeNIe` AND `GeNIe-Ada`

In this section, we provide further details on the computational complexity of `GeNIe` across multiple noising ratios $r$ and `GeNIe-Ada` when operating on a search space of $r \in [0.6, 0.8]$. Computational complexity has been reported in terms of the total number of inference/denoising-diffusion steps and the runtime in seconds per generated image. The runtime has been averaged over 10 different image-generations on an NVIDIA Tesla-V100 GPU with 16GB VRAM with 50 steps of denoising using a DPM scheduler with StableDiffusion

Table A4: Computational Complexity

| **Augmentation** | **Inf. Steps** | **Runtime [sec/img]** |
|---|---|---|
| `Txt2Img` | $T$ | 4.12 |
| `GeNIe(r=0.5)` | $0.5 \times T$ | 2.17 |
| `GeNIe(r=0.6)` | $0.6 \times T$ | 2.59 |
| `GeNIe(r=0.7)` | $0.7 \times T$ | 2.98 |
| `GeNIe(r=0.8)` | $0.8 \times T$ | 3.46 |
| `GeNIe-Ada` | $2.1 \times T$ | 9.22 |

v1.5. As can be seen in Tab. A4, `GeNIe` is approximately $1/r$ times faster than the base diffusion model (referred to as the `Txt2Img` augmentation baseline). This empirically corroborates with the total number of denoising steps using in `GeNIe` vs. `Txt2Img`. Since, `GeNIe-Ada` scans for the best hard-negative in $r \in [0.6, 0.8]$, it incurs a computational cost of $\approx 2.2\times$ the `Txt2Img`. Note that the runtime for `GeNIe-Ada` reported in Tab. A4 also includes the runtime of performing a batched forward pass through a ResNet-50 feature extraction backbone.

### A.8 DETAILS OF FINE-GRAINED FEW-SHOT CLASSIFICATION

Here we provide details of Fine-grained Few-shot Classification experiments.

**Datasets:** We assess our method on several datasets: Food101 (Bossard et al., 2014) with 101 classes of foods, CUB200 (Wah et al., 2011) with 200 bird species classes, Cars196 (Krause et al., 2013) with 196 car model classes, and FGVC-Aircraft (Maji et al., 2013) with 41 aircraft manufacturer classes. We provide detailed information around fine-grained datasets in Table A5. The reported metric is the average Top-1 accuracy over 100 episodes. Each episode involves sampling 20 classes and 1-shot from the training set, with the final model evaluated on the respective test set.

**Implementation Details:** We enhance the basic prompt by incorporating the superclass name for the fine-grained dataset: "A photo of a <target class>, a type of <superclass>". For instance, in the *food* dataset and the *burger* class, our prompt reads: "A photo of a *burger*, a type of *food*." No additional augmentation is used for generative methods in this context. We generate 19 samples for both cases of our method and also the baseline with weak augmentation.

Table A5: Train and test split details of the fine-grained datasets. We use the provided train set for few-shot task generation, and the provided test sets for our evaluation. Aircraft dataset uses the manufacturer hierarchy.

| Dataset | Classes | Train samples | Test samples |
|---|---|---|---|
| CUB200 (Wah et al., 2011) | 200 | 5994 | 5794 |
| Food101 (Bossard et al., 2014) | 101 | 75750 | 25250 |
| Cars (Krause et al., 2013) | 196 | 8144 | 8041 |
| Aircraft (Maji et al., 2013) | 41 | 6,667 | 3333 |

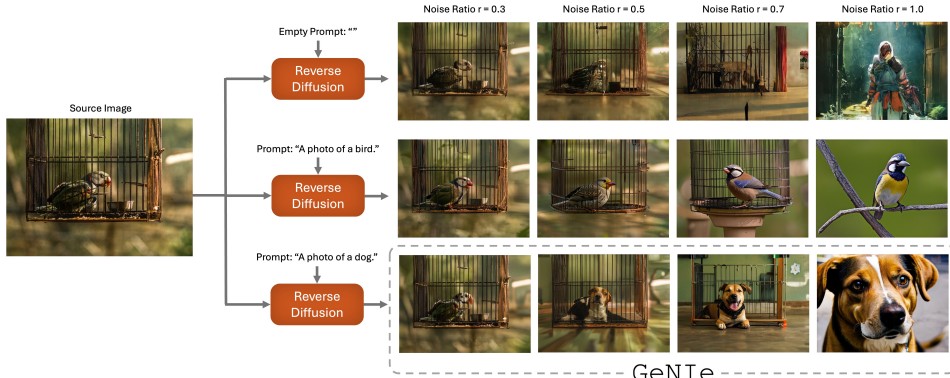

Figure A4: **Key components of `GeNIe`:** (i) careful choice of $r$ and (ii) contradictory prompt are two key idea behind `GeNIe`

## A.9 FEW-SHOT CLASSIFICATION WITH RESNET-34 ON *tiered*IMAGENET

Table A6: *tiered*-**ImageNet**: Accuracies (% ± std) for 5-way, 1-shot and 5-way, 5-shot classification settings on the test-set. We compare against various SOTA supervised and unsupervised few-shot classification baselines as well as other augmentation methods, with UniSiam (Lu et al., 2022) pre-trained ResNet-34 backbone.

| | ResNet-34 | | | |
| --- | --- | --- | --- | --- |
| **Augmentation** | **Method** | **Pre-training** | **1-shot** | **5-shot** |
| Weak | MAML + dist (Finn et al., 2017) | sup. | 51.7±1.8 | 70.3±1.7 |
| Weak | ProtoNet (Snell et al., 2017) | sup. | 52.0±1.2 | 72.1±1.5 |
| Weak | UniSiam + dist (Lu et al., 2022) | unsup. | 68.7±0.4 | 85.7±0.3 |
| Weak | UniSiam (Lu et al., 2022) | unsup. | 65.0±0.7 | 82.5±0.5 |
| Strong | UniSiam (Lu et al., 2022) | unsup. | 64.8±0.7 | 82.4±0.5 |
| CutMix (Yun et al., 2019) | UniSiam (Lu et al., 2022) | unsup. | 63.8±0.7 | 80.3±0.6 |
| MixUp (Zhang et al., 2018) | UniSiam (Lu et al., 2022) | unsup. | 64.1±0.7 | 80.0±0.6 |
| Img2Img$^L$(Luzi et al., 2022) | UniSiam (Lu et al., 2022) | unsup. | 66.1±0.7 | 83.1±0.5 |
| Img2Img$^H$(Luzi et al., 2022) | UniSiam (Lu et al., 2022) | unsup. | 70.4±0.7 | 84.7±0.5 |
| Txt2Img(He et al., 2022b) | UniSiam (Lu et al., 2022) | unsup. | 75.0±0.6 | 85.4±0.4 |
| DAFusion (Trabucco et al., 2024) | UniSiam (Lu et al., 2022) | unsup. | 64.1±2.1 | 82.8±1.4 |
| GeNIe (Ours) | UniSiam (Lu et al., 2022) | unsup. | **75.7±0.6** | **86.0±0.4** |
| GeNIe-Ada (Ours) | UniSiam (Lu et al., 2022) | unsup. | **76.9±0.6** | **86.3±0.2** |

We follow the same evaluation protocol here as mentioned in section 4.1. As summarized in Table A6, `GeNIe` and `GeNIe-Ada` outperform all other data augmentation techniques.

## A.10 ADDITIONAL DETAILS OF LONG-TAIL EXPERIMENTS

We present a comprehensive version of Table 4 to benchmark the performance with different backbone architectures (e.g., ResNet50) and to compare against previous long-tail baselines; this is detailed in Table A8.

**Implementation Details of LViT:** We download the pre-trained ViT-B of LViT (Xu et al., 2023) and finetune it with Bal-BCE loss proposed therein on the augmented dataset. Training takes 2 hours on four NVIDIA RTX 3090 GPUs. We use the same hyperparameters as in (Xu et al., 2023) for finetuning: 100 epochs, $lr = 0.008$, batch size of 1024, CutMix and MixUp for the data augmentation.

**Implementation Details of VL-LTR:** We use the official code of VL-LTR (Tian et al., 2022) for our experiments. We use a pre-trained CLIP ResNet-50 backbone. We followed the hyperparameters reported in VL-LTR (Tian et al., 2022). We augment only "Few" category and train the backbone with the VL-LTR (Tian et al., 2022) method. Training takes 4 hours on 8 NVIDIA RTX 3090 GPUs.

## A.11 EXTRA COMPUTATION OF GeNIe-ADA

Given that GeNIe-Ada searches for the best hard-negative between multiple noise-ratios $r$'s, it naturally requires a higher compute budget than `txt2Img` that only uses $r = 1$. For this experiment, we

Table A7: Few-shot classification comparison of GeNIe-Ada with `Txt2Img` on miniImagenet.

| Method | ResNet-18 | | ResNet-34 | | ResNet-50 | |
|---|---|---|---|---|---|---|
| | 1-shot | 5-shot | 1-shot | 5-shot | 1-shot | 5-shot |
| `Txt2Img` | 76.9±1.0 | 86.5±0.9 | 77.1±0.8 | 86.7±1.0 | 77.2±1.3 | 86.8±0.9 |
| `GeNIe-Ada` | 77.7±0.8 | 87.4±1.0 | 78.3±0.9 | 87.8±0.9 | 79.1±1.1 | 88.4±1.2 |

use GeNIe-Ada with $r \in \{0.6, 0.7, 0.8\}$ to compare with `Txt2Img`. Based on this, we only have 3 paths, with steps of 0.1), and for each of which we go through partial reverse diffusion process. E.g. for $r = 0.6$ we do 30 steps instead of standard 50 steps of Stable Diffusion. This practically breaks down the total run-time of `GeNIe-Ada` to approximately 2 times that of the standard reverse diffusion (GeNIe-Ada: total $r = 0.6 + 0.7 + 0.8 = 2.1$ vs `Txt2Img` total $r = 1$). Thus, to be fair, we generate twice as many `Txt2Img` augmentations as compared to GeNIe-Ada to keep a constant compute budget across the methods, following your suggestion. The results are shown in Table A7. As can be seen, even in this new setting, GeNIe-Ada offers a performance improvement of $0.8\%$ to $1.9\%$ across different backbones.

Table A8: **Long-Tailed ImageNet-LT:** We compare different augmentation methods on ImageNet-LT and report Top-1 accuracy for "Few", "Medium", and "Many" sets. † indicates results with ResNeXt50. ∗: indicates training with 384 resolution so is not directly comparable with other methods with 224 resolution. On the "Few" set and LiVT method, our augmentations improve the accuracy by 11.7 points compared to LiVT original augmentation and 4.4 points compared to `Txt2Img`.

| Method | Many | Med. | Few | Overall Acc |
|---|---|---|---|---|
| **ResNet-50** | | | | |
| CE (Cui et al., 2019) | 64.0 | 33.8 | 5.8 | 41.6 |
| LDAM (Cao et al., 2019) | 60.4 | 46.9 | 30.7 | 49.8 |
| c-RT (Kang et al., 2020) | 61.8 | 46.2 | 27.3 | 49.6 |
| $\tau$-Norm (Kang et al., 2020) | 59.1 | 46.9 | 30.7 | 49.4 |
| Causal (Tang et al., 2020) | 62.7 | 48.8 | 31.6 | 51.8 |
| Logit Adj. (Menon et al., 2021) | 61.1 | 47.5 | 27.6 | 50.1 |
| RIDE(4E)† (Wang et al., 2021) | 68.3 | 53.5 | 35.9 | 56.8 |
| MiSLAS (Zhong et al., 2021) | 62.9 | 50.7 | 34.3 | 52.7 |
| DisAlign (Zhang et al., 2021a) | 61.3 | 52.2 | 31.4 | 52.9 |
| ACE† (Cai et al., 2021) | 71.7 | 54.6 | 23.5 | 56.6 |
| PaCo† (Cui et al., 2021a) | 68.0 | 56.4 | 37.2 | 58.2 |
| TADE† (Zhang et al., 2021b) | 66.5 | **57.0** | 43.5 | 58.8 |
| TSC (Li et al., 2022f) | 63.5 | 49.7 | 30.4 | 52.4 |
| GCL (Li et al., 2022e) | 63.0 | 52.7 | 37.1 | 54.5 |
| TLC (Li et al., 2022a) | 68.9 | 55.7 | 40.8 | 55.1 |
| BCL† (Zhu et al., 2022) | 67.6 | 54.6 | 36.6 | 57.2 |
| NCL (Li et al., 2022c) | 67.3 | 55.4 | 39.0 | 57.7 |
| SAFA (Hong et al., 2022) | 63.8 | 49.9 | 33.4 | 53.1 |
| DOC (Wang et al., 2022) | 65.1 | 52.8 | 34.2 | 55.0 |
| DLSA (Xu et al., 2022) | 67.8 | 54.5 | 38.8 | 57.5 |
| ResLT (Cui et al., 2022) | 63.3 | 53.3 | 40.3 | 55.1 |
| PaCo (Cui et al., 2021b) | 68.2 | 58.7 | 41.0 | 60.0 |
| LWS (Kang et al., 2019) | 62.2 | 48.6 | 31.8 | 51.5 |
| Zero-shot CLIP (Radford et al., 2021) | 60.8 | 59.3 | 58.6 | 59.8 |
| DRO-LT (Samuel & Chechik, 2021) | 64.0 | 49.8 | 33.1 | 53.5 |
| VL-LTR (Tian et al., 2022) | 77.8 | 67.0 | 50.8 | 70.1 |
| Cap2Aug (Roy et al., 2023) | 78.5 | **67.7** | 51.9 | 70.9 |
| `GeNIe-Ada` | **79.2** | 64.6 | **59.5** | **71.5** |
| **ViT-B** | | | | |
| LiVT∗ (Xu et al., 2023) | 76.4 | 59.7 | 42.7 | 63.8 |
| ViT (Dosovitskiy et al., 2021) | 50.5 | 23.5 | 6.9 | 31.6 |
| MAE (He et al., 2022a) | 74.7 | 48.2 | 19.4 | 54.5 |
| DeiT (Touvron et al., 2022) | 70.4 | 40.9 | 12.8 | 48.4 |
| LiVT (Xu et al., 2023) | 73.6 | 56.4 | 41.0 | 60.9 |
| LiVT + `Img2Img`$^L$ | 74.3 | 56.4 | 34.3 | 60.5 |
| LiVT + `Img2Img`$^H$ | 73.8 | 56.4 | 45.3 | 61.6 |
| LiVT + `Txt2Img` | **74.9** | 55.6 | 48.3 | 62.2 |
| LiVT + `GeNIe` (r=0.8) | 74.5 | 56.7 | 50.9 | 62.8 |
| LiVT + `GeNIe-Ada` | 74.0 | **56.9** | **52.7** | **63.1** |

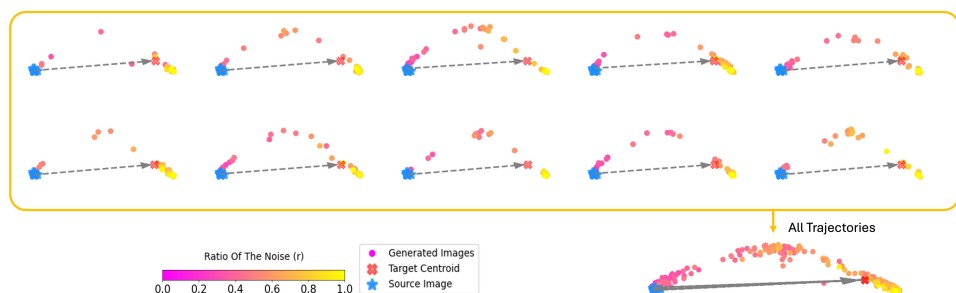

Figure A5: Analyzing the semantic trajectory of `GeNIe` augmentations across 10 different images of class Mushroom (source image) to class Volcano (target class).

### A.12 FURTHER ANALYSIS OF SEMANTIC SHIFTS USING GENIE

In Fig. 5, we empirically demonstrate that by increasing the noise ratio from 0 to 1, the semantic category of the source image transitions gradually from the source class to the text-prompt's target class. To establish this further, we now choose 10 samples of a source class of Mushroom and generate `GeNIe` augmentations with the target class of a Volcano. The generated images corresponding to each $r \in [0, 1]$ are passed through a DINOv2 encoder and their embeddings are projected onto their 2 principle eigen vectors using PCA. The trajectories extracted from each of these 10 source images is depicted collectively and individually in Fig. A5. It can be noticed that each of the trajectories demonstrate a gradual transition of semantic category from the source to the target class, with a sparse distribution of points usually observed within $[0.4, 0.6]$. This is also observed in the plot on the bottom-right side of the figure where all trajectories are collectively plotted. Here, however, there is no clear range of $r$ where a sparse distribution of points can be observed, thus indicating that each source image has its own optimal $r$ value. This can be attributed to the inter-sample variances of images belonging to the same class. Since `GeNIe-Ada` operates on each individual source image and target class text-prompt, it facilitates the selection of the best hard-negative per sample.

### A.13 HOW DOES GENIE CONTROL WHICH FEATURES ARE RETAINED OR CHANGED?

We instruct the diffusion model to generate an image by combining the latent noise of the source image with the textual prompt of the target category. This combination is controlled by the amount of added noise and the number of reverse diffusion iterations. This approach aims to produce an image that aligns closely with the semantics of the target category while preserving the background and features from the source image that are unrelated to the target.

To demonstrate this, in Figure A4, We are progessivley moving towards the two key components of GeNIe: (i) careful choice of $r$ and (ii) contradictory prompt. The input image is a bird in a cage. The top row shows a Stable Diffusion model, unprompted. As can be seen, such a model can generate anything (irrespective of the input image) with a large $r$. Now prompting the same model with "a photo of a bird" allows the model to preserve low-level and contextual features of the input image (up to $r = 0.7$ and 0.8), until for a large $r \geq 0.9$ it returns a bird but the context has nothing to do with the source input. This illustrates how a careful choice of $r$ can help preserve such low-level features, and is a key idea behind GeNIe. However, we also need a semantic switch to a different target class as shown in the last row where a hardly seen image of a dog in a cage is generated by a combination of a careful choice of $r$ and the contradictory prompt - leading to the full mechanics of GeNIe. This sample now serves as hard negative for the source image (bird class).

### A.14 ANALYZING NOISE EFFECTS IN BI-DIRECTIONAL TRANSFORMATIONS WITH GENIE

To further explore the effect of noise ratio $r$ in `GeNIe`, we conducted an experiment where `GeNIe` was applied twice to transform between a source image and a target category. For this experiment, images from the "mushroom" category were used as the source, and "volcano" served as the target category. In the first step, we applied `GeNIe` using a mushroom image as the source and a volcano prompt as the target. In the second step, we reversed the process: the `GeNIe`-generated volcano

image from the first step was used as the source, with the target prompt set to mushroom. Importantly, using a smaller noise ratio, $r$ during the generation of the volcano image helps preserve more low-level visual features from the original mushroom source image. Consequently, when the roles of source and target are flipped in the second step, the final image retains a stronger resemblance to the original mushroom source image for lower noise ratios. This phenomenon is visualized in Fig. A6. As shown, a lower noise ratio during the first step results in the preservation of more visual features, leading to a final image that more closely resembles the original mushroom source.

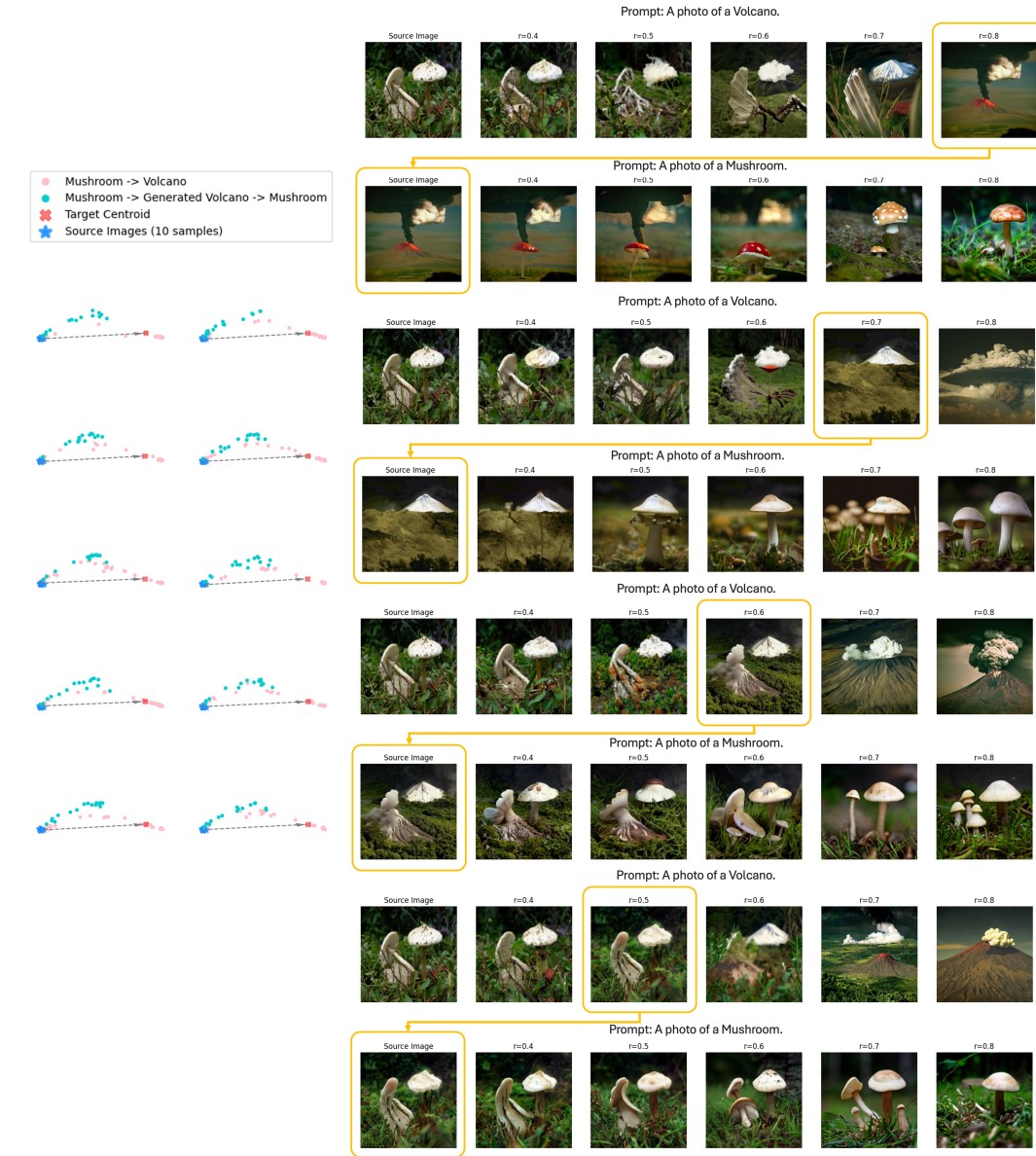

Figure A6: **Trajectory of GeNIe augmentations:** To further analyze the effect of noise ratio $r$ in GeNIe, we conducted an experiment using a set of augmentations generated from 10 different source images in the "mushroom" category, with a target label of "Volcano," across varying noise ratios. Similar to Fig. 5, all generated augmentations were processed through the DinoV2 ViT-G model, which serves as our oracle, to extract their embeddings. For visualization, we applied PCA to these embeddings. Next, we selected one augmentation with a specific noise ratio, ($r$), and used it as the source image in for the "volcano" category in GeNIe, with the target prompt set to "mushroom." As observed, using a lower noise ratio samples as the source for "volcano" preserves more low-level visual features from the original mushroom source image. Consequently, after a second round of applying GeNIe, the resulting augmentations (even rows) tend to more closely resemble the original source image (first image in the corresponding odd rows above). The left plot presents the embeddings of all 10 samples, while the right plot provides a detailed visualization of one sample, showcasing the impact of varying noise ratios used in the second step of applying GeNIe.

A.15 MORE VISUALIZATIONS

Additional qualitative results resembling the style presented in Fig. 4 are presented in Fig. A8, and more visuals akin to Fig. 2 can be found in Fig. A9. Moreover, we also present more visualization similar to the style in Fig. 5 in Fig. A7.

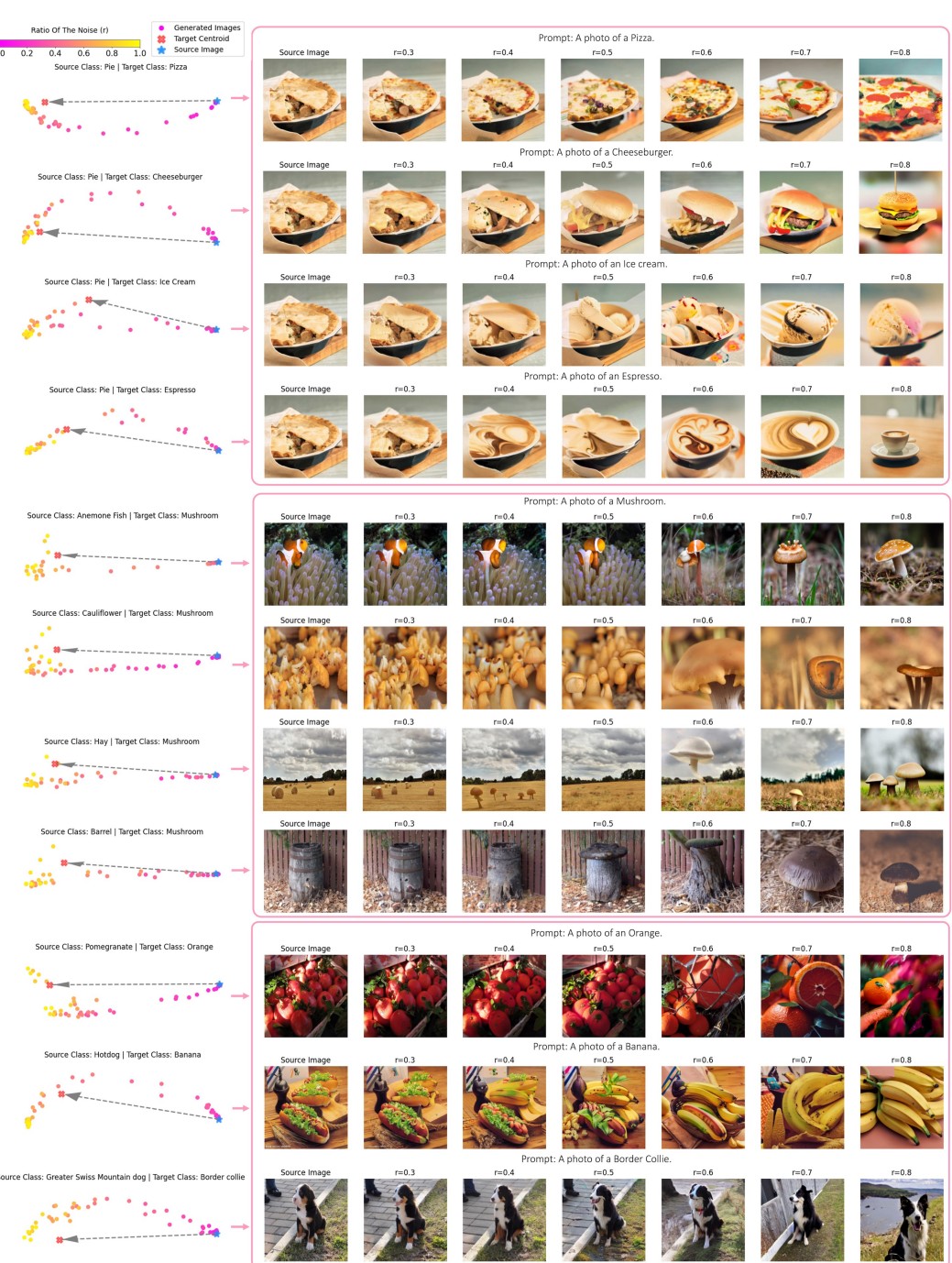

Figure A7: **Effect of noise in GeNIe:** Similar to Fig. 5, we pass all the generated augmentations through the DinoV2 ViT-G model, which acts as our oracle model, to obtain their associated embeddings. Subsequently, we employ PCA for visualization purposes. The visualization reveals that the magnitude of semantic transformations is contingent upon both the source image and the specified target category.

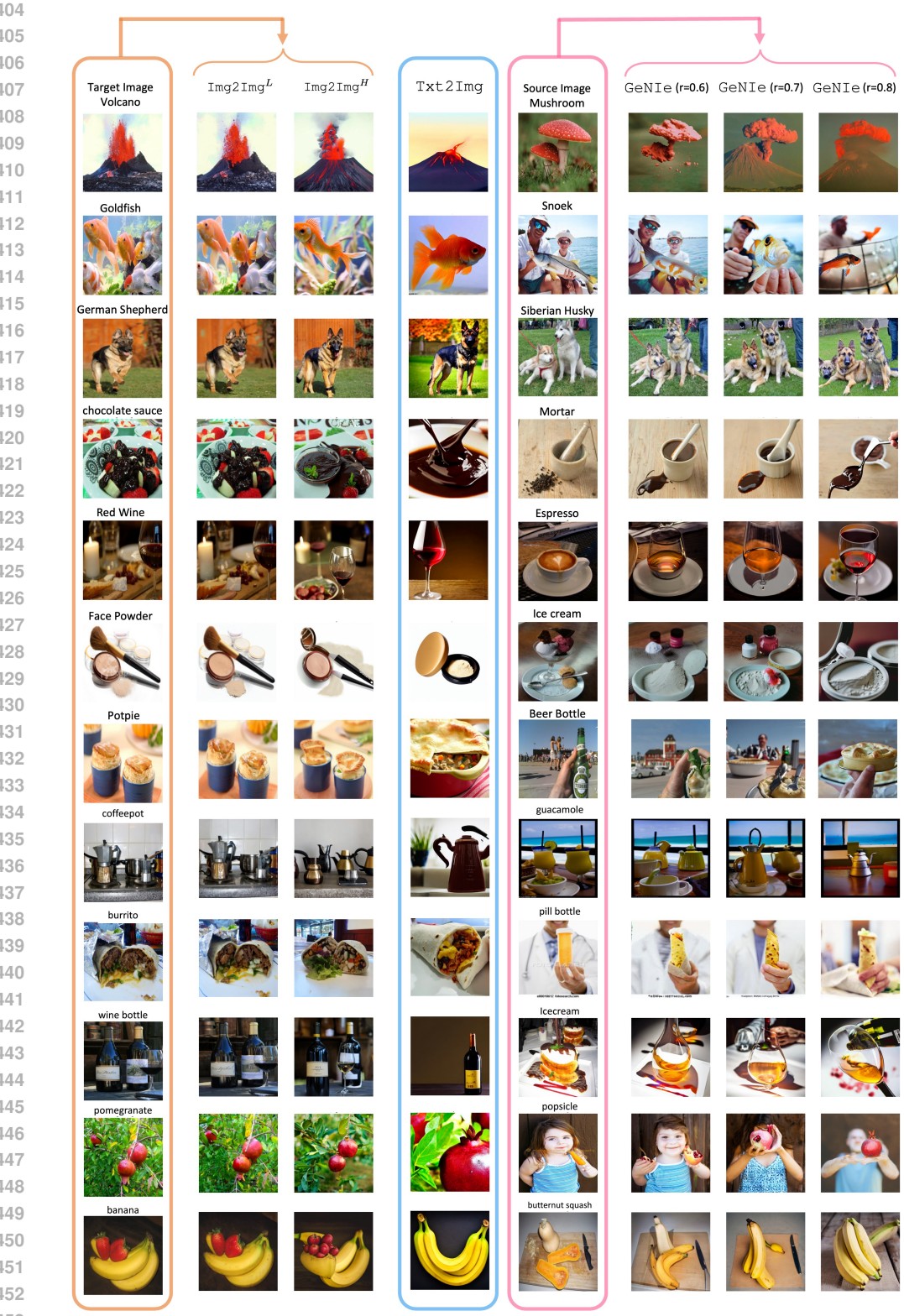

Figure A8: **Visualization of Generative Samples:** More visualization akin to Fig. 4. We compare GeNIe with two baselines: **Img2Img$^L$ augmentation** uses both image and text prompt from the same category, resulting in less challenging examples. **Txt2Img augmentation** generates images based solely on a text prompt, potentially deviating from the task's visual domain. **GeNIe augmentation** incorporates the target category name in the text prompt along with the source image, producing desired images with an optimal amount of noise, and balancing the impact of the source image and text prompt.

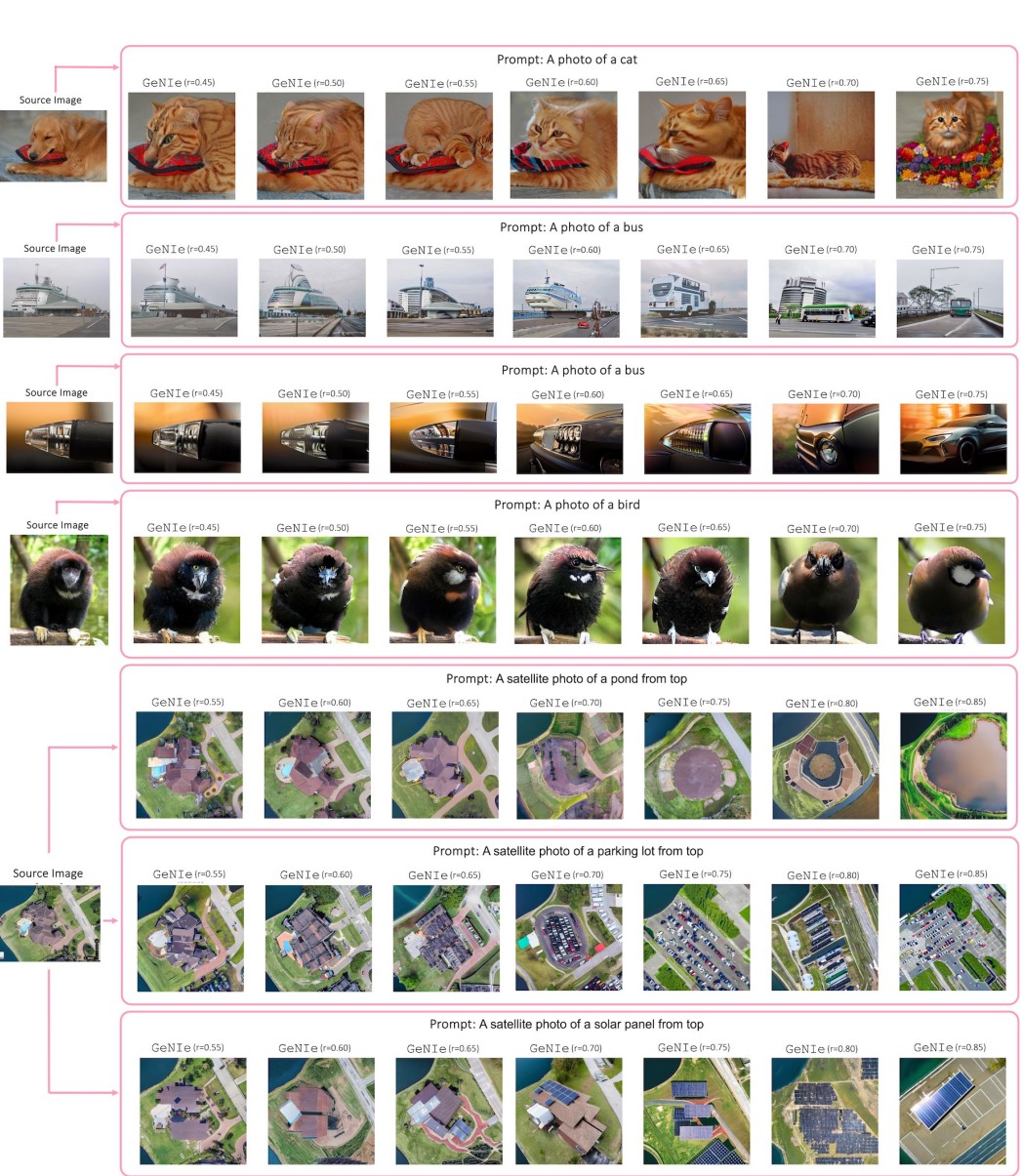

Figure A9: **Effect of noise in GeNIe:** Akin to Fig. 2, we use GeNIe to create augmentations with varying noise levels. As is illustrated in the examples above, a reduced amount of noise leads to images closely mirroring the semantics of the source images, causing a misalignment with the intended target label.

