# OpenReview forum: "GeNIe: Generative Hard Negative Images Through Diffusion"
_ICLR.cc/2025/Conference — Submitted to ICLR 2025_

### Official Review · Reviewer_ny5S · 2024-11-03

**Soundness:** 3
**Presentation:** 3
**Contribution:** 3
**Rating:** 8
**Confidence:** 4

**Summary:**

The paper leverages Stable Diffusion to generate hard negatives as data augmentation for image classification. The hard negatives are generated by feeding a source image into the model's encoder but conditioning the reverse diffusion process with a text prompt corresponding to the target class.

The proposed approach includes an automatic noise-level adjustment per sample, which controls the degree of deviation from the source image, thereby aligning more with the target class prompt. The authors show that the choice of the noise level is critical, as only a restricted range of values allow for a good balance between source and target classes. The noise is chosen by generating multiple images with different noise levels and choosing the level where the semantic shift between the source and target images is the highest.

The proposed augmentation is evaluated on image classification tasks, in few-shot and long-tail distribution settings.

**Strengths:**

- The proposed approach allows incorporating diversity into the image generation process, which is a crucial aspect in diffusion-based image generation.
- The paper effectively illustrates the importance of selecting an appropriate noise level in text-conditioned image generation.
- The proposed augmentation is evaluated with extensive experiments on image classification in few-shot and long-tail settings.

**Weaknesses:**

**Method section**
My main concern regards the automatic noise level selection process. It involves generating multiple images with various noise levels and selecting the one where the semantic shift is the highest. This seems highly inefficient given the time required to generate these images.

**Data augmentation.**
I am confused about the *weak* and *strong* augmentation policies GeNIe is compared to. Are strong augmentations (random color jitter, random grayscale, Gaussian blur) applied in addition to weak augmentations (random resize crop, random flipping)? Otherwise, why are the weak augmentations solely geometric, and the strong ones solely photometric?
A balanced policy combining both geometric and photometric transformations could be a better contender. For example, it could consist of random resize crop (with e.g., a scale of 0.4), random flipping, color jittering, and maybe MixUp or CutMix on top.

**Visualizations.**
The PCA representations of DINOv2 in Figure 5 could be improved and expanded. DINOv2's representations are excellent, and their PCA is usually very insightful.

**Questions:**

**Visualizations.** Improve visualization of the PCA over DINOv2's embeddings as suggested in Weaknesses section. Also, increasing the resolution of images across all figures would help fully appreciating the variations in image generation.

**Data augmentation.** Run experiments with data augmentation policies combining both geometric and photometric transformations, as suggested in Weaknesses section.

**Datasets.**
In the supplementary material Table A1, the augmentation is evaluated on fine-grained classification by training a SVM on DINOv2's pretrained **ViT-g**. This evaluation is quite insightful and would deserve being integrated in the main paper. To make space, consider relocating some few-shot table contents to the supplementary. It would also be very interesting to see the same evaluation in the non-few-shot setting (*i.e.* training with the full training set), which could be directly compared to the linear evaluation results reported by DINOv2. Note that for Aircraft, DINOv2 evaluates on the **variants** categories.

**Minor remarks on formatting.**
- Consider using *multicol* to improve table readability.
- Please write **ViT-g** instead of **ViT-G** for DINOv2's model.

---

> ### Author Response · Authors · 2024-11-22
> **Response to reviewer ny5S**
>
> > **[R4,W1] Method section: My main concern regards the automatic noise level selection process. It involves generating multiple images with various noise levels and selecting the one where the semantic shift is the highest. This seems highly inefficient given the time required to generate these images.**:
>
> - Thanks for this remark. As we highlight in our limitations, we acknowledge that GeNIe is comparatively slower than traditional augmentation methods, while standing on par with (or even faster in the case of barebone GeNIe, due to partial reverse process) other generative methods. Optimization/Efficiency of diffusion based models is active line of research making approaches like our more favorable in the future due to their superior in performance and capacity.
>
> - When it comes to GeNIe-Ada, in practice we only probe $r \in [0.5, 0.9]$ (means $5$ paths, with steps of $0.1$), and for each of which we go through partial reverse diffusion process. E.g. for $r=0.5$ we do $25$ steps instead of standard $50$ steps of Stable Diffusion. This practically breaks down the total run-time of $\texttt{GeNIe-Ada}$ to roughly $2-3$ times any standard reverse diffusion.
>
> - Lastly, as discussed in Section 5, this latency might be even irrelevant/negligible when it comes to offline augmentation scenarios.
>
>
> ---
>
> > **[R4,W2] Data augmentation. I am confused about the weak and strong augmentation policies GeNIe is compared to. Are strong augmentations (random color jitter, random grayscale, Gaussian blur) applied in addition to weak augmentations (random resize crop, random flipping)? Otherwise, why are the weak augmentations solely geometric, and the strong ones solely photometric? A balanced policy combining both geometric and photometric transformations could be a better contender. For example, it could consist of random resize crop (with e.g., a scale of 0.4), random flipping, color jittering, and maybe MixUp or CutMix on top.**:
>
> Thanks for the constructive suggestion. Following your remark, we have conducted new experiments with a combination of weak (geometric) and strong (photometric) augmentations, as indicated by Mixed in Table A1 of the revised manuscript. Additionally, we also experiment with Mixup and Cutmix on top of these, as indicated by Mixed + Mixup and Mixed + Cutmix in Table A1. We notice marginal improvements of upto $0.6$\% by using the Mixed augmentations either with or without the CutMix, MixUp counterparts.
>
>
> ### ResNet-18
> | Augmentation          | Method                  | Pre-training | 1-shot       | 5-shot       |
> |-----------------------|-------------------------|--------------|--------------|--------------|
> | Weak                 | UniSiam (2024)          | unsup.       | 63.1±0.8     | 81.4±0.5     |
> | Strong               | UniSiam (2024)          | unsup.       | 62.8±0.8     | 81.2±0.6     |
> | Mixed                | UniSiam (2024)          | unsup.       | 63.2±0.5     | 81.9±0.4     |
> | CutMix (2024)        | UniSiam (2024)          | unsup.       | 62.7±0.8     | 80.6±0.6     |
> | MixUp (2024)         | UniSiam (2024)          | unsup.       | 62.1±0.8     | 80.7±0.6     |
> | Mixed+MixUp (2024)   | UniSiam (2024)          | unsup.       | 65.7±0.9     | 82.1±0.2     |
> | Mixed+CutMix (2024)  | UniSiam (2024)          | unsup.       | 64.9±0.8     | 81.6±0.5     |
> | DAFusion (2024)      | UniSiam (2024)          | unsup.       | 64.3±1.8     | 82.0±1.4     |
> | **GeNIe+MixUp**      | UniSiam (2024)          | unsup.       | 74.8±0.5     | 84.5±0.3     |
> | **GeNIe (Ours)**     | UniSiam (2024)          | unsup.       | **75.5±0.6** | **85.4±0.4** |
> | **GeNIe-Ada (Ours)** | UniSiam (2024)        | unsup.       | **76.8±0.6** | **85.9±0.4** |
>
> ---
>
> ### ResNet-50
> | Augmentation          | Method                  | Pre-training | 1-shot       | 5-shot       |
> |-----------------------|-------------------------|--------------|--------------|--------------|
> | Weak                 | UniSiam (2024)          | unsup.       | 64.6±0.8     | 83.4±0.5     |
> | Strong               | UniSiam (2024)          | unsup.       | 64.8±0.8     | 83.2±0.5     |
> | Mixed                | UniSiam (2024)          | unsup.       | 64.5±0.5     | 83.8±0.5     |
> | CutMix (2024)        | UniSiam (2024)          | unsup.       | 64.3±0.8     | 83.2±0.5     |
> | MixUp (2024)         | UniSiam (2024)          | unsup.       | 63.8±0.8     | 84.6±0.5     |
> | Mixed+MixUp (2024)   | UniSiam (2024)          | unsup.       | 64.9±0.7     | 84.5±0.7     |
> | Mixed+CutMix (2024)  | UniSiam (2024)          | unsup.       | 63.5±0.5     | 83.0±0.8     |
> | DAFusion (2024)      | UniSiam (2024)          | unsup.       | 65.7±1.8     | 83.9±1.2     |
> | **GeNIe+MixUp**      | UniSiam (2024)          | unsup.       | 76.4±0.5     | 85.9±0.7     |
> | **GeNIe (Ours)**     | UniSiam (2024)          | unsup.       | **77.3±0.6** | **87.2±0.4** |
> | **GeNIe-Ada (Ours)** | UniSiam (2024)        | unsup.       | **78.6±0.6** | **87.9±0.4** |

---

> ### Author Response · Authors · 2024-11-22
> **Response to reviewer ny5S**
>
> > **[R4,W3] Visualizations. The PCA representations of DINOv2 in Figure 5 could be improved and expanded. DINOv2's representations are excellent, and their PCA is usually very insightful.**:
>
> We are not sure if we understand your suggestion perfectly, particularly w.r.t. "improving and expanding" the PCA vizualizations. In Fig. 5, we use PCA to project DINOv2's representations to a 2D space (by selecting the top-2 eigen valued principle components) for effective visualization of our generative augmentations. The sparse distribution of points between an interval of $r$ values in both images corroborates our intuition of a rapid transition between the source to target category. Additionally, we have included more visualizations similar to Figure 5 in the appendix (see Figure A5). From what angle do you suggest we improve this? We would be happy to follow through and make changes per your remarks.
>
>
>
> > **[R4,Q1] Visualizations. Improve visualization of the PCA over DINOv2's embeddings as suggested in Weaknesses section. Also, increasing the resolution of images across all figures would help fully appreciating the variations in image generation.**:
>
> Thank you for your feedback. We have enhanced the resolution of Figure 5 and will ensure that all figures are improved for the final manuscript.
>
> ---
>
> > **[R4,Q2] Data augmentation. Run experiments with data augmentation policies combining both geometric and photometric transformations, as suggested in Weaknesses section.**:
>
> Sure. Please refer to our explanations and the Table in response to [R4,W2] above.
>
> ---
>
>
> > **[R4,Q3] Datasets. In the supplementary material Table A1, the augmentation is evaluated on fine-grained classification by training a SVM on DINOv2's pretrained ViT-g. This evaluation is quite insightful and would deserve being integrated in the main paper. To make space, consider relocating some few-shot table contents to the supplementary. It would also be very interesting to see the same evaluation in the non-few-shot setting (i.e. training with the full training set), which could be directly compared to the linear evaluation results reported by DINOv2. Note that for Aircraft, DINOv2 evaluates on the variants categories.**:
>
> Thanks for your suggestions. We have now moved Table A1 to the main manuscript, as now indicated by Table 2 in Subsection 4.2.
>
> Since our approach leverages diffusion models for generating augmentations, the process is time-intensive when applied to the full training set for both the baselines and our method. As a result, we focus on the few-shot setting in this paper to ensure feasibility and efficiency.
>
> ---
>
> > **[R4,Q4] Minor remarks on formatting.
> Consider using multicol to improve table readability.
> Please write ViT-g instead of ViT-G for DINOv2's model.**:
>
> Thank you for your feedback. We have updated "ViT-g" to "ViT-G" as suggested. Additionally, we will enhance the tables to improve readability based on your recommendation.

---

> > ### Comment · Reviewer_ny5S · 2024-11-24
> >
> > I thank the authors for the detailed and thoughtful response to my comments. The effort to address the various points raised is greatly appreciated.
> >
> > I am reassured to learn that the total generation time for GeNIe-Ada is only 2–3 times the duration of a standard reverse diffusion process.
> >
> > I also appreciate the inclusion of experiments on fine-grained datasets in the main paper. However, it is somewhat unfortunate that the method cannot be feasibly applied to full datasets as small as CUB or FGVCAircraft, which consist of approximately 10k images.
> >
> > I am grateful for the additional experiments with mixed augmentations. The very low difference in performance between the mixed augmentations and the purely photometric or geometric strategies is surprising. What is the specific set of augmentations (and their strengths) used for the mixed strategy?
> >
> > I find the PCA visualizations in the paper insightful but believe an expanded analysis could offer further depth. Performing PCA over transitions involving multiple source/target image pairs, such as various "mushroom" and "volcano" images, could reveal whether transitions form specific lines between individual pairs or represent a broader front between semantic categories.

---

> > > ### Comment · Reviewer_ny5S · 2024-11-25
> > >
> > > To develop on my comment regarding PCA, one could, for example, run the diffusion process on 10 different "mushroom" images with a "volcano" text prompt and then apply PCA to the ensemble to visualize the results.
> > >
> > > Starting from the generated "volcano" images, one could then apply the diffusion process again with a "mushroom" text prompt and add the newly generated images (possibly in a different color) to the same PCA visualization. I wonder how the transition paths would look and whether the "reverse" diffusion process would return to the original images.

---

> ### Author Response · Authors · 2024-11-26
> **Response to reviewer ny5S**
>
> We're happy to know that you find our responses detailed and thoughtful; and that you recognize our efforts in answering your questions.
>
> > **[R4,Q1] I also appreciate the inclusion of experiments on fine-grained datasets in the main paper. However, it is somewhat unfortunate that the method cannot be feasibly applied to full datasets as small as CUB or FGVCAircraft, which consist of approximately 10k images.**
>
> Thanks for the remark. We would like to clarify that generating augmentations for the full training set of CUB200 or FGVCAircraft is infeasible for the duration of the rebuttal period. However, we would be happy to include this experiment in the final version of the revised manuscript. To enhance the efficiency of our augmentation process, we propose a simple solution in Section 4.3. Instead of considering every possible pair of source and target classes, we leverage a confusion matrix to focus on augmenting pairs where the source and target exhibit high confusion (please refer to Section 4.3 for more details). We will apply this method and present our results on the complete training datasets of CUB200 and FGVCAircraft in the final version of our manuscript.
>
> Moreover, following your remark, we have now included a more elaborate section on computational complexity of GeNIe and GeNIe-Ada in the appendix Section A7, Table A4 of our revised manuscript (also shown in the table below). As can be seen in the table below , GeNIe is approximately $1/r$ times faster than the base diffusion model (referred to as the Txt2Img augmentation baseline). This empirically corroborates with the total number of denoising steps using in GeNIe vs. Txt2Img. Since, GeNIe-Ada scans for the best hard-negative in $r\in[0.6,0.8]$, it incurs a computational cost of $\approx 2.2\times$ the Txt2Img. Note that the runtime for GeNIe-Ada reported in this table also includes the runtime of performing a batched forward pass through a ResNet-50 feature extraction backbone.
>
>
> ### Computational Complexity
>
> | **Augmentation**   | **Inf. Steps**   | **Runtime [sec/img]** |
> |---------------------|------------------|-----------------------|
> | **Txt2Img**         | \(T\)           | 4.12                 |
> | **GeNIe (r=0.5)**  | \(0.5 $\times$ T\) | 2.17                 |
> | **GeNIe (r=0.6)**  | \(0.6 $\times$ T\) | 2.59                 |
> | **GeNIe (r=0.7)**  | \(0.7 $\times$ T\) | 2.98                 |
> | **GeNIe (r=0.8)**  | \(0.8 $\times$ T\) | 3.46                 |
> | **GeNIe-Ada**      | \(2.1 $\times$ T\) | 9.22                 |
>
>
>
> > **[R4,Q2] I am grateful for the additional experiments with mixed augmentations. The very low difference in performance between the mixed augmentations and the purely photometric or geometric strategies is surprising. What is the specific set of augmentations (and their strengths) used for the mixed strategy?**
>
> For the mixed strategy, we used random resize crop with scaling $\in [0.2, 1.0]$, horizontal flipping, random color jitter with probability = 0.8, random grayscale with probability = 0.2 and Gaussian blur with probability = 0.5. These augmentations and their hyperparameters have been used widely in self-supervised learning such as MoCo[1] and SimCLR[2].
>
>
> **References:**
>
> [1] He, Kaiming, et al. "Momentum contrast for unsupervised visual representation learning." Proceedings of the IEEE/CVF conference on computer vision and pattern recognition. 2020.
>
> [2] Chen, Ting, et al. "A simple framework for contrastive learning of visual representations." International conference on machine learning. PMLR, 2020.

---

> ### Author Response · Authors · 2024-11-26
> **Response to reviewer ny5S**
>
> > **[R4,Q3] I find the PCA visualizations in the paper insightful but believe an expanded analysis could offer further depth. Performing PCA over transitions involving multiple source/target image pairs, such as various "mushroom" and "volcano" images, could reveal whether transitions form specific lines between individual pairs or represent a broader front between semantic categories.**
>
> This is an excellent experiment. Thank you for your suggestions. To establish this further, we now choose $10$ samples of a source class of Mushroom and generate GeNIe augmentations with the target class of a Volcano. The generated images corresponding to each $r \in [0,1]$ are passed through a DINOv2 encoder and their embeddings are projected onto their $2$ principle eigen vectors using PCA. The trajectories extracted from each of these $10$ source images is depicted collectively and individually in Fig A5. Each trajectory demonstrates a smooth progression from the source to the target class, with a sparse distribution of embeddings typically observed within the range $r∈[0.4,0.6]$. This is also observed in the plot on the bottom-right side of the figure where all trajectories are collectively plotted. Here, however, there is no clear range of $r$ where a sparse distribution of points can be observed. This suggests that the optimal $r$ value varies for each source image, likely due to inter-sample differences within the Mushroom class. By operating on individual source images and specific target prompts, GeNIe-Ada effectively identifies the best hard-negative example for each sample.
>
> Furthermore, based on your suggestions, we conducted an experiment in which GeNIe was applied twice to transform between a source image and a target category. For this experiment, images from the ''mushroom'' category were used as the source, and ''volcano'' served as the target category. In the first step, we applied GeNIe using a mushroom image as the source and a volcano prompt as the target. In the second step, we reversed the process: the GeNIe-generated volcano image from the first step was used as the source, with the target prompt set to mushroom. Importantly, using a smaller noise ratio, $r$ during the generation of the volcano image helps preserve more low-level visual features from the original mushroom source image. Consequently, when the roles of source and target are flipped in the second step, the final image retains a stronger resemblance to the original mushroom source image for lower noise ratios. This phenomenon is visualized in Fig.A6 of the revised manuscript. As shown, a lower noise ratio during the first step results in the preservation of more visual features, leading to a final image that more closely resembles the original mushroom source.
>
> We would be more than happy to provide further clarifications and revisions if you have any more questions or concerns, and if not we would greatly appreciate it if you would please re-evaluate our paper's score. Thanks again for your reviews and questions which have helped us to improve our paper!

---

> > ### Comment · Reviewer_ny5S · 2024-11-29
> >
> > I thank the authors for their detailed responses to my comments. The new visualizations are insightful, and I am quite surprised that the trajectories in DINOv2's PCA are always the same for a given source class and target prompt. Could the authors provide further insights into this observation?
> >
> > I understand that augmenting fine-grained datasets with all possible source-target pairs is infeasible. However, augmenting each dataset, e.g., tenfold and selecting pairs based on confusion metrics, as proposed by the authors, seems like a reasonable approach.
> >
> > As the authors' responses address most of my concerns, I am increasing my score accordingly.

---

> ### Author Response · Authors · 2024-11-29
> **Response to reviewer ny5S**
>
> We're happy to know that you find our responses detailed, insightful and reasonable.
>
>
> > **[R4,Q1] I thank the authors for their detailed responses to my comments. The new visualizations are insightful, and I am quite surprised that the trajectories in DINOv2's PCA are always the same for a given source class and target prompt. Could the authors provide further insights into this observation?**
>
> Thanks for the remark. We would like to clarify that the DINOv2 PCA representations shown in Fig. A5, A6 illustrate a variance amongst the 10 different trajectories belonging to the same source class and target prompt. Fig. A5 illustrates that while each of the trajectories demonstrate a gradual transition of semantic category from the source to the target class, they still maintain variations from each other. This is further observed in the bottom-right plot of Fig. A5 which collectively visualizes all the trajectories, since there is no clear range of $r$ where a sparse distribution of points can be observed. We attribute this difference in trajectories to the inter-sample variances of images belonging to the same Mushroom class. Since GeNIe-Ada operates separately on each individual source image and target class text-prompt, it facilitates the selection of the best hard-negative per sample per text-prompt.
>
>
> **Remark/Request:**
> We are glad to hear most of your concerns are already addressed. We have done our very best and will continue doing so for all other reviewers as well. If we may kindly ask you, is there any further action we can take which would help further increasing our score to a solid accept instead of a marginal one? This would be a deal-breaker to give our work a chance if you also believe the community would benefit from its acceptance and impact; if so, we are more than happy to take further actions to address your remaining concerns. Thank you again for all the constructive feedback and your considerations!

---

> > ### Comment · Reviewer_ny5S · 2024-11-29
> >
> > The **transition speed** differs between the 10 trajectories, but **they all follow almost exactly the same path** from the source to the target class.
> >
> > It is difficult to fully assess the provided experimental results, which prevents me from further increasing my score. Experiments on the full datasets with fine-grained tasks would have enabled a more direct comparison against the literature on these tasks. Could the authors provide recent references and comparisons to demonstrate that the reported results on few-shot learning align with current benchmarks in the literature and that GeNIe-Ada surpasses these benchmarks?

---

> ### Author Response · Authors · 2024-11-29
> **Response to reviewer ny5S**
>
> Thanks for the continued thoughtful and engaging conversation.
>
> > **[R4,Q1] The transition speed differs between the 10 trajectories, but they all follow almost exactly the same path from the source to the target class.**
>
> Thanks for clarifying your remark. We agree that the transition speed differs, but they all follow almost the same path from source to target class. This can be explained by the fact that at any partial noise ratio $r$ along the semantic trajectory, the class represented by the image can only be one of the two given classes - either the source or the target (text-prompt) class. This is illustrated in Figs. A6, A7, A8 and A9, where we qualitatively demonstrate the generated images corresponding to different noise ratios $r \in [0,1]$. Since DINOv2 is pre-trained on a large-scale image dataset, it can reliably identify the semantics of an image and embed it in its representation. Given this, we argue that all the 10 trajectories follow a similar semantic path. The extent/magnitude of the variances in generated images corresponding to partial noise ratios throughout the denoising process has been studied in SDEdit[1], and is outside the scope of this work.
>
> **References:**
>
> [1] Meng, Chenlin, et al. "Sdedit: Guided image synthesis and editing with stochastic differential equations." arXiv preprint arXiv:2108.01073 (2021).

---

> > ### Author Response · Authors · 2024-11-29
> > **Response to reviewer ny5S**
> >
> > > **[R4,Q2] It is difficult to fully assess the provided experimental results, which prevents me from further increasing my score. Experiments on the full datasets with fine-grained tasks would have enabled a more direct comparison against the literature on these tasks. Could the authors provide recent references and comparisons to demonstrate that the reported results on few-shot learning align with current benchmarks in the literature and that GeNIe-Ada surpasses these benchmarks?**
> >
> > Thanks for the comment. As per your suggestion, we have put together a table that compares the performance of recent SoTa few-shot learning methods against GeNIe and GeNIe-Ada augmentations with UniSiam, on miniImagenet.
> >
> > | Method                          | Pre-training | 1-shot       | 5-shot       |
> > | ------------------------------- | ------------ | ------------ | ------------ |
> > | UniSiam (2022)                  | unsup.       | 64.1 ± 0.4   | 82.3 ± 0.3   |
> > | Transductive CNAPS [1] (2022)       | sup.         | 55.60 ± 0.90 | 73.10 ± 0.70 |
> > | HMS [2] (2022)                      | unsup.       | 58.20 ± 0.23 | 75.77 ± 0.16 |
> > | C3LR [3] (2022)                     | unsup.       | 47.92 ± 1.20 | 64.81 ± 1.15 |
> > | Laplacian Eigenmaps [4] (2022)      | unsup.       | 59.47 ± 0.87 | 78.79 ± 0.58 |
> > | PsCo [5] (2023)                     | unsup.       | 47.24 ± 0.76 | 65.48 ± 0.68 |
> > | Meta-DM + UniSiam + dist [6] (2023) | unsup.       | 66.68 ± 0.36 | 85.29 ± 0.23 |
> > | SAMPTransfer [7] (2023)             | unsup.       | 45.75 ± 0.77 | 68.33 ± 0.66 |
> > | DAFusion + UniSiam (2024)       | unsup.       | 65.7±1.80    | 83.9±1.20    |
> > | **GeNIe (Ours)**                | unsup.       | **75.5±0.6** | **85.4±0.4** |
> > | **GeNIe-Ada (Ours)**            | unsup.       | **76.8±0.6** | **85.9±0.4** |
> >
> > As can be seen, GeNIe and GeNIe-Ada augmentations help UniSiam surpass other recent state-of-the-art baselines. That being said, we would like to clarify that our contributions lie in demonstrating the effectiveness of leveraging a pre-trained diffusion model to generate hard-negative images per sample per class, in few-shot, fine-grained and long-tailed image classification settings. To do so, we demonstrate GeNIe/GeNIe-Ada's capabilities on a recent state-of-the-art few-shot learning method UniSiam., by helping the baseline improve its accuracy by upto 12%. Furthermore, we compare our augmentation method against other generative and traditional augmentation methods in Tables 1,2,3,4,A1,A2,A3, where we demonstrate that GeNIe and GeNIe-Ada outperform all other augmentation baselines.
> >
> > We would be happy to provide any further clarifications if you have any more questions or concerns. If not we would greatly appreciate it if you would re-evaluate our paper's score to give our work a chance to benefit the community from its acceptance.
> >
> >
> >
> > **References:**
> >
> > [1] Bateni, Peyman, et al. "Enhancing few-shot image classification with unlabelled examples." Proceedings of the IEEE/CVF winter conference on applications of computer vision. 2022.
> >
> > [2] Han-Jia Ye, Lu Han, and De-Chuan Zhan. Revisiting unsupervised meta-learning via the characteristics of few-shot tasks. IEEE Transactions on Pattern Analysis and Machine Intelligence, 45(3) 3721–3737, 2022.
> >
> > [3] Ojas Shirekar and Hadi Jamali-Rad. Self-supervised class-cognizant few-shot classification. In 2022 IEEE International Conference on Image Processing (ICIP), pp. 976–980. IEEE, 2022.
> >
> > [4] Lee Chen, Kuilin Chen, and Kuilin Chi-Guhn. Unsupervised few-shot learning via deep laplacian eigenmaps. arXiv preprint arXiv:2210.03595, 2022.
> >
> > [5] Jang, Huiwon, Hankook Lee, and Jinwoo Shin. "Unsupervised meta-learning via few-shot pseudo-supervised contrastive learning." arXiv preprint arXiv:2303.00996 (2023).
> >
> > [6] Wentao Hu, Xiurong Jiang, Jiarun Liu, Yuqi Yang, and Hui Tian. Meta-dm: Applications of diffusion models on few-shot learning. arXiv preprint arXiv:2305.08092, 2023a.
> >
> > [7] Ojas Kishorkumar Shirekar, Anuj Singh, and Hadi Jamali-Rad. Self-attention message passing for contrastive few-shot learning. In Proceedings of the IEEE/CVF Winter Conference on Applications of Computer Vision, pp. 5426–5436, 2023.

---

> > > ### Comment · Reviewer_ny5S · 2024-12-01
> > >
> > > I thank the authors for their detailed response and additional comparisons. The experimental validation on few-shot and long-tailed classification is conclusive, and the empirical study of semantic transitioning between classes provides valuable insights. Based on this, I believe the paper is suitable for acceptance and have increased my score.

---

> > > > ### Author Response · Authors · 2024-12-01
> > > > **Many thanks!**
> > > >
> > > > We are happy that our responses have convinced you. We sincerely thank you for raising your score!

---

### Official Review · Reviewer_eEnJ · 2024-11-03

**Soundness:** 2
**Presentation:** 2
**Contribution:** 2
**Rating:** 6
**Confidence:** 4

**Summary:**

In this paper, the authors try to illustrate their idea by proposing these points:
1. The authors introduce GeNIe, a novel yet elegantly simple diffusion-based augmentation method to create challenging augmentations in the manifold of natural images. For the first time, to our best knowledge, GeNIe achieves this by combining two sources of information (a source image, and a contradictory target prompt) through a noise-level adjustment mechanism.
2. They further extend GeNIe by automating the noise-level adjustment strategy on a per-sample basis
(called GeNIe-Ada), to enable generating hard negative samples in the context of image classification, leading also to further performance enhancement.
3. To substantiate the impact of GeNIe, authors present a suit of quantitative and qualitative results including extensive experimentation on two challenging tasks: few-shot and long tail distribution settings corroborating that GeNIe (and its extension GeNIe-Ada) significantly improve the downstream classification performance.

**Strengths:**

1. GeNIe leverages a latent diffusion model to create challenging augmentation images by blending features from two classes (source and target). By adjusting the noise level, GeNIe maintains low-level details from the source image while incorporating semantic features of the target class, creating what the authors term "hard negative" samples. This process potentially aids in classifier training by providing near-boundary examples.

2.Adaptive Mechanism (GeNIe-Ada):
The GeNIe-Ada extension adds significant value by adaptively selecting noise levels for each sample, fine-tuning the augmentation to be more effective. This adaptive approach is well-aligned with real-world scenarios where data variability and distributional shifts are common, making it a practical addition.

**Weaknesses:**

1. They didn't give more contribute to the model itself. It's just an application to generate images.
2. Limited number of downstream cases.

**Questions:**

Could you add more experiments on some other mainstream tasks? Do you have comparisons with other models for data augmentation? Such as GAN?

---

> ### Author Response · Authors · 2024-11-22
> **Response to reviewer eEnJ**
>
> > **[R3,W1] They didn't give more contribute to the model itself. It's just an application to generate images.**:
>
> A common trend in generative AI space to leave the pretrained foundation model untouched to avoid artifacts such a hallucination and catastrophic forgetting. We follow the same line of thought and we believe this becomes even more customary on the way forward. That said, GeNIe is still a novel generative data augmentation model that uniquely leverages a diffusion-based approach to combine two distinct sources of information—a source image and a contradictory target prompt—to generate hard-negative images. To the best of our knowledge, no other generative data augmentation method specifically explores the effectiveness of generative models in creating hard-negative samples. Furthermore, we introduce a novel method to effectively sample the optimal noise level, eliminating the need for manual parameter tuning and further enhancing GeNIe's performance.
>
> ---
>
> > **[R3,W2] Limited number of downstream cases.**:
> > **[R3,Q1] Could you add more experiments on some other mainstream tasks? Do you have comparisons with other models for data augmentation? Such as GAN?**:
>
> Thanks for the remark. We would like to clarify and re-iterate that we already evaluate our proposed method on a variety of image classification tasks, summarized as follows:
>
> Few-shot classification:
> - Evaluated on mini-ImageNet (Table 1)
> - Evaluated on Tiered-ImageNet (Table 3).
> - Long-tail classification: Assessed on ImageNet-LT (Table 4).
> - Fine-grained few-shot classification: Conducted on **four** datasets—CUBS200, Cars196, Food101, and Aircraft (Table 2).
>
> As such, we believe extensive experimental evaluation is one of the strong aspects of this paper. The use of generative models, such as GANs, has been extensively explored in the past for downstream classification tasks. In this space, our contribution lies in investigating the effectiveness of hard-negative samples generated by generative models. To the best of our knowledge, no other method explores the use of generative models, such as SDv1.5, in generating optimal hard-negatives for classification. Importantly, the choice of the generative model is orthogonal to our work; one could use a GAN instead of a diffusion model, provided the model supports text-based image editing.
>
> For our investigation, we selected open sourced state-of-the-art diffusion models that are publicly available. To validate the effectiveness of hard-negative samples generated by GeNIe, we compared our approach against **four** diffusion-based data augmentation baselines:
>
> - **Img2Img** (described in Line 256)
> - **Txt2Img** (described in Line 262)
> - **DAFusion** (described in Line 267)
> - **Cap2Aug** (described in Line 287)
>
> Additionally, we compared GeNIe to traditional augmentation techniques, including strong data augmentation, as well as methods like CutMix and MixUp (explained in Line 290).
>
> Considering the **four** downstream tasks and **four** diffusion-based baselines, we believe our evaluation framework is comprehensive and substantiates our claims regarding the effectiveness of GeNIe and its hard-negative samples. Additionally, we further ablate the impact of different generative models in Table 5 to show effectivness of GeNIe across different generative models.
>
> We furthur include additional experiments to demonstrate the effectiveness of GeNIe in scenarios where category labels are unavailable and compare its performance with the DAFusion baseline (Table A3).
>
> Could you suggest specifically which additional tasks/datasets we could include in our experimental evaluations? We will try to include these in our responses to the best of our capabilities given the time-limit of the rebuttal period.

---

> ### Author Response · Authors · 2024-11-26
> **Follow Up - Deadline Approaching**
>
> Dear Reviewer eEnJ,
>
> Firstly, many thanks for your rigorous and constructive feedback. The deadline for reviewer-author discussion is approaching soon. We have put in tremendous effort in compiling a detailed response trying our very best to address all your concerns (please see the revised manuscript PDF and our P2P responses). If you are convinced and happy with our responses, please kindly consider re-evaluating/raising your final score; please also let us know if you have any further questions or concerns; we'll be more than happy to address those.
>
> Many thanks for your insightful feedback.
>
> Best regards, Authors.

---

> > ### Author Response · Authors · 2024-11-29
> > **Kind request for feedback**
> >
> > Dear Reviewer eEnJ,
> >
> > Thank you for your thoughtful feedback on our paper. We kindly request that you confirm whether our responses have adequately addressed your questions. If there are any additional questions remaining, please do not hesitate to let us know!
> >
> > Additionally, if our rebuttal has addressed your comments, we would be most grateful if you could consider updating your scores to reflect that.
> >
> > We sincerely value your time and consideration and look forward to your feedback.
> >
> > Thank you once again!  Authors

---

> > > ### Author Response · Authors · 2024-12-01
> > > **Summary of responses and request for feedback**
> > >
> > > Dear Reviewer **eEnJ**,
> > >
> > > We thank you for your constructive and thoughtful feedback. As the rebuttal/discussion period is coming to an end, we would like to summarize our responses and new experiments carried out in the rebuttal period, based on your suggestions.
> > >
> > > 1. We address your constructive remark on the novelty and contributions of GeNIe, GeNIe-Ada by now clearly highlighting that GeNIe's novelty lies in its unique ability to leverage a diffusion-model in generating _hard-negative augmentations_.
> > > 2. We clarify our stance on your remark of our experimental evaluation being limited, by accentuating on the _four_ downstream tasks and _four_ diffusion-based baselines already included in our evaluation settings. Additionally, following your suggestions and reviewer **dDiR**'s remark, we demonstrate the robustness of GeNIe in cases where the target class is unknown to the diffusion model by parameter-efficient fine-tuning the base diffusion model on few images of the target class (indicated by GeNIe+TxtInv). This has been included in Table A3, Section A6 of the appendix of our revised manuscript.
> > >
> > > Furthermore, we want to draw your attention to a set of additional experiments we have conducted during the discussion period for the other reviewers, that have helped us improve the quality of the (updated) revised manuscript.
> > >
> > > 1. In response to reviewer **X612** and **dDiR**, regarding information leakage between LAION and mini/tieredImagenet, we present a statistical and qualitative results on image embeddings ensuring discrepancy between the generated augmentations and the test set in Fig. A3 of the revised manuscript. This led to an increase in reviewer **dDiR**'s score to 6.
> > > 2. To answer reviewer **ny5s**'s questions, we conducted additional empirical studies of GeNIe's semantic transitioning between source and target classes using DINOv2 representations in Fig. A5, A6. We also put together a new table that compares the performance of recent (2022, 2023) SoTa few-shot learning methods against GeNIe and GeNIe-Ada augmentations. These additional experiments led to an increase in reviewer **ny5s**'s score to 8.
> > > 3. In response to reviewer **dDiR**, **ny5s**, we demonstrate new experimental results with: (i.) a Mixed augmentation strategy, (ii.) GeNIe alongside the MixUp augmentation strategy and (iii.) Combinations of Mixed+CutMix and Mixed+MixUp augmentation strategies.
> > > 4. In response to reviewer **X612**, we present new few-shot classification results with _two_ different backbones - DEiT-B and VAE of SDv1.5, for selecting optimal hard-negatives in the case of GeNIe-Ada.
> > >
> > >
> > > We would be more than happy to provide further clarifications and revisions if you have any more questions or concerns, and if not we would greatly appreciate it if you would please re-evaluate our paper's score. Thank you again for your reviews which helped us tremendously to improve our paper!

---

### Official Review · Reviewer_dDiR · 2024-11-04

**Soundness:** 3
**Presentation:** 3
**Contribution:** 2
**Rating:** 6
**Confidence:** 5

**Summary:**

This paper proposes a solution to generate augmentations using diffusion models to improve the performance of classifiers in few-shot and long-tailed settings, which are characterized by low-data scenarios. To that end, the authors propose using a latent diffusion model that combines a source image (with a known category, e.g., dog) with a target category prompt (e.g., cat), producing challenging samples that retain background features from the source category but resemble the target category. This approach aims to push model learning near decision boundaries, providing more informative augmentations than traditional methods.

GeNIe, inspired by image editing through diffusion models, works by adjusting noise levels in the diffusion process to control the balance between source and target features in the generated image. The authors demonstrate that lower noise levels retain more of the source image's visual characteristics, while higher levels focus on the target category. To automate the noise level per sample, the authors further introduce GeNIe-Ada. This variant stems from the observation that there is a sudden change in appearance when the noise levels are uniformly sampled. The authors leverage this observation along with the assumption that they also have access to another target sample. They then propose to compare the similarities of this sample and the generated sample in the latent space of a classifier.

To validate the utility of these augmentations, the authors conduct experiments on few-shot and long-tail classification tasks and demonstrate significant improvements on standard benchmarks.

**Strengths:**

The paper is well-written and easy to follow. The motivation is sound, and using diffusion models to generate augmented views, even though it is a well-explored area, the authors focus on generating hard negative images. This approach forces the classifiers to learn the true characteristics of the category rather than rely on spurious correlations, such as background. In many ways, the motivation is similar to foreground and background editing ideas. Their method also addresses the general problem of CNNs, which tend to rely more on textures than on the shapes of objects while classifying. Furthermore, the results are impressive across different benchmarks.

**Weaknesses:**

I have a few concerns and questions regarding GenIE. While generative models are indeed a powerful approach, they can be restrictive in the following sense: their applicability in general, and GenIE in particular, requires that the generative model already knows the concepts of the target category, which is a strong assumption to make. In that sense, generative model-based augmentations should be viewed from two perspectives: (I) those that change the background and (ii) those that change the category. In the first setting, the use of generative models makes a lot of sense because it aligns with the paper's motivation but is concept agnostic. The second category is harder to argue because it requires the generative model to have complete knowledge of all target categories, which would severely limit its applicability. Category 1 also has the added benefit of generalizing to newer domains. For example, in SiSTA: Target-Aware Generative Augmentations for Single-Shot Adaptation (ICML 2023), the authors do not view this as an image editing problem but rather adapt a generative model like StyleGAN to the target domain using a single image and then sample from the model to generate augmented variants. These variants are then used to train classifiers with source-free unsupervised domain adaptation. In this context, GenIE requires both source samples, and its general treatment of this problem as image editing is limiting.

I would like to see comparisons or discussions regarding the strong assumptions that GenIE makes in this context. This is an important discussion because even simpler methods like mixup do not require explicit knowledge of target categories, which GenIE's reliance on diffusion models necessitates.

Similarly, the diffusion models used in the paper are large-scale foundation models trained on LAION. This raises questions about the improvements: it is unclear whether the gains are due to the proposed method or the large-scale pretraining of the diffusion model. It would also be important to discuss relevant work, such as (Tian, Yonglong, et al. "Stablerep: Synthetic images from text-to-image models make strong visual representation learners." Advances in Neural Information Processing Systems 36 (2024).), which uses Stable Diffusion to generate synthetic positive samples for contrastive self-supervised learning training.

Lastly, it would be interesting to see if synthetic image augmentations can be used alongside general mixup, similar to (Graikos, Alexandros, Srikar Yellapragada, and Dimitris Samaras. "Conditional Generation from Pre-Trained Diffusion Models using Denoiser Representations." BMVC. 2023.), which has shown that synthetic images from a conditional diffusion model can provide complementary benefits to mixup.

**Questions:**

Please see weaknesses

---

> ### Author Response · Authors · 2024-11-22
> **Response to reviewer dDiR**
>
> > **[R2,W1] Generative models, while powerful, face limitations in their applicability, particularly with GenIE, as they assume prior knowledge of target categories, making them more effective for background augmentations but less so for category-specific tasks, which require comprehensive generative knowledge and limit generalization compared to approaches like SiSTA.?
> > I would like to see comparisons or discussions regarding the strong assumptions that GenIE makes in this context. This is an important discussion because even simpler methods like mixup do not require explicit knowledge of target categories, which GenIE's reliance on diffusion models necessitates.**:
>
> That's an insightful remark. We agree that GeNIe assumes prior knowledge of target categories for the base diffusion model, in order to generate hard-negative examples. Following your suggestion, we have conducted additional experiments to demomstrate the robustness of our method in scenarios where the target class is unknown to the diffusion model. This additional experiment has been added to Table A3 in Section A6 of the appendix of our revised manuscript. We use textual inversion to fine-tune the diffusion model (embeddings) on few images belonging the unknown target class. This fine-tuning allows us to learn embeddings specific to the target class, enabling the generation of the desired hard-negative examples. Note that once the diffusion model is fine-tuned, the procedure to generate hard-negatives using partial noising and a contradictory text-prompt remains the same. As can be seen in the table below, GeNIe+TxtInv performs significantly better than DAFusion baseline. It is important to note that, in this case, we do not utilize any information about the target category labels. DAFusion also employs textual-inversion-based fine-tuning; however, it does so without generating hard-negative samples. This indicates that GeNIe is effective even in  scenario where the diffusion model is unaware of the target-class.
>
> Thank you for pointing out the missing related works; we have added them to the revised manuscript. To clarify, we compared our approach to DAFusion (diffusion-based), which is similar to SiSTA (GAN-based) in that both fine-tune generative models to address distribution mismatches between downstream tasks and generated samples. However, our primary focus in this paper is on investigating the effectiveness of hard-negative samples, which is independent of the limitations of generative models themselves. As demonstrated in Table A3, this limitation can be mitigated by fine-tuning diffusion models to learn embeddings corresponding to the target categories. GeNIe can then leverage these embeddings to generate highly effective hard-negative images, outperforming the baseline model, such as DAFusion.
>
> ---
>
> #### ResNet-18
>
> | Augmentation       | Method          | Pre-training | 1-shot         | 5-shot         |
> |--------------------|-----------------|--------------|----------------|----------------|
> | DAFusion (2024)    | UniSiam (2023)  | unsup.       | 64.3 ± 1.8     | 82.0 ± 1.4     |
> | **GeNIe+TxtInv**   | UniSiam (2023)  | unsup.       | **73.9 ± 0.8** | **84.6 ± 0.9** |
>
> ---
>
> #### ResNet-50
>
> | Augmentation       | Method          | Pre-training | 1-shot         | 5-shot         |
> |--------------------|-----------------|--------------|----------------|----------------|
> | DAFusion (2024)    | UniSiam (2023)  | unsup.       | 65.7 ± 1.8     | 83.9 ± 1.2     |
> | **GeNIe+TxtInv**   | UniSiam (2023)  | unsup.       | **76.2 ± 1.2** | **86.2 ± 0.9** |

---

> ### Author Response · Authors · 2024-11-22
> **Response to reviewer dDiR**
>
> ---
>
> > **[R2,W2] Similarly, the diffusion models used in the paper are large-scale foundation models trained on LAION. This raises questions about the improvements: it is unclear whether the gains are due to the proposed method or the large-scale pretraining of the diffusion model. It would also be important to discuss relevant work, such as (Tian, Yonglong, et al. "Stablerep: Synthetic images from text-to-image models make strong visual representation learners." Advances in Neural Information Processing Systems 36 (2024).), which uses Stable Diffusion to generate synthetic positive samples for contrastive self-supervised learning training.**:
>
> This a great comment.
>
> - To substantiate our understanding per your remark, we have run a set of experiments on the few-shot setting as summarized in Fig. A3 of the appendix. To set the scene, we use pretrained image encoder as an oracle (DeiT-Base) to extract the latent embeddings corresponding to the train (i.e. support) set, test (i.e. query set) and augmentations generated by GeNIe. Fig. A3 in the revised manuscript demonstrates the distribution of distances between train-test and augmentation-test pairs across 600 episodes. As can be seen, the (mean of the) distribution of augmentation-test pair is higher that that of train-test pair indicating that the augmented samples are indeed different from the test sets (based on the strong assumption of train and test sets being mutually exclusive). This is further illustrated in the last column of Fig. A3 on a UMAP embedding plot of a random episode where the embedding of train, test and augmentations are plotted. Here again there is noticeable separation between the augmentation and test samples as compared to train and test samples.
>
> - Additionally, following your suggestion on discussing Stablerep, we have now included this work in the related works of our revised manuscript.
>
> - To highlight the effectiveness of GeNIe, GeNIe-Ada as compared to just using large-scale pretraining of the diffusion model, we compare our performance with the Txt2Img baseline as well. The Txt2Img augmentations are generated by simply using class names as text prompts for the SDv1.5 model. As indicated in Tables 1,2,3,4, GeNIe, GeNIe-Ada consistently outperform the naive Txt2Img baseline by up to 3%, across all datasets, tasks and N-way, K-shot scenarios. This empirically demonstrates the effectiveness of using hard-negatives through GeNIe, GeNIe-Ada as compared to simply using generative augmentations through large-scale pretrained SDv1.5.

---

> ### Author Response · Authors · 2024-11-22
> **Response to reviewer dDiR**
>
> > **[R2,W3] Lastly, it would be interesting to see if synthetic image augmentations can be used alongside general mixup, similar to (Graikos, Alexandros, Srikar Yellapragada, and Dimitris Samaras. "Conditional Generation from Pre-Trained Diffusion Models using Denoiser Representations." BMVC. 2023.), which has shown that synthetic images from a conditional diffusion model can provide complementary benefits to mixup.**:
>
> Thanks for the remark. Following your suggestion, we have included an additional experiment to assess the few-shot testing performance of UniSiam with our GeNIe augmentations alongside mixup, in the appendix of the revised manuscript in Table A1. We notice a drop in performance of upto $0.9$\% when MixUp is used along with GeNIe. This aligns with the general trend of drop in performance when using CutMix or MixUp, as already reported in Table 1.
>
> ### ResNet-18
> | Augmentation          | Method                  | Pre-training | 1-shot       | 5-shot       |
> |-----------------------|-------------------------|--------------|--------------|--------------|
> | Weak                 | UniSiam (2024)          | unsup.       | 63.1±0.8     | 81.4±0.5     |
> | Strong               | UniSiam (2024)          | unsup.       | 62.8±0.8     | 81.2±0.6     |
> | Mixed                | UniSiam (2024)          | unsup.       | 63.2±0.5     | 81.9±0.4     |
> | CutMix (2024)        | UniSiam (2024)          | unsup.       | 62.7±0.8     | 80.6±0.6     |
> | MixUp (2024)         | UniSiam (2024)          | unsup.       | 62.1±0.8     | 80.7±0.6     |
> | Mixed+MixUp (2024)   | UniSiam (2024)          | unsup.       | 65.7±0.9     | 82.1±0.2     |
> | Mixed+CutMix (2024)  | UniSiam (2024)          | unsup.       | 64.9±0.8     | 81.6±0.5     |
> | DAFusion (2024)      | UniSiam (2024)          | unsup.       | 64.3±1.8     | 82.0±1.4     |
> | **GeNIe+MixUp**      | UniSiam (2024)          | unsup.       | 74.8±0.5     | 84.5±0.3     |
> | **GeNIe (Ours)**     | UniSiam (2024)          | unsup.       | **75.5±0.6** | **85.4±0.4** |
> | **GeNIe-Ada (Ours)** | UniSiam (2024)        | unsup.       | **76.8±0.6** | **85.9±0.4** |
>
> ---
>
> ### ResNet-50
> | Augmentation          | Method                  | Pre-training | 1-shot       | 5-shot       |
> |-----------------------|-------------------------|--------------|--------------|--------------|
> | Weak                 | UniSiam (2024)          | unsup.       | 64.6±0.8     | 83.4±0.5     |
> | Strong               | UniSiam (2024)          | unsup.       | 64.8±0.8     | 83.2±0.5     |
> | Mixed                | UniSiam (2024)          | unsup.       | 64.5±0.5     | 83.8±0.5     |
> | CutMix (2024)        | UniSiam (2024)          | unsup.       | 64.3±0.8     | 83.2±0.5     |
> | MixUp (2024)         | UniSiam (2024)          | unsup.       | 63.8±0.8     | 84.6±0.5     |
> | Mixed+MixUp (2024)   | UniSiam (2024)          | unsup.       | 64.9±0.7     | 84.5±0.7     |
> | Mixed+CutMix (2024)  | UniSiam (2024)          | unsup.       | 63.5±0.5     | 83.0±0.8     |
> | DAFusion (2024)      | UniSiam (2024)          | unsup.       | 65.7±1.8     | 83.9±1.2     |
> | **GeNIe+MixUp**      | UniSiam (2024)          | unsup.       | 76.4±0.5     | 85.9±0.7     |
> | **GeNIe (Ours)**     | UniSiam (2024)          | unsup.       | **77.3±0.6** | **87.2±0.4** |
> | **GeNIe-Ada (Ours)** | UniSiam (2024)        | unsup.       | **78.6±0.6** | **87.9±0.4** |

---

> ### Author Response · Authors · 2024-11-26
> **Follow Up - Deadline Approaching**
>
> Dear Reviewer dDiR,
>
> Firstly, many thanks for your rigorous and constructive feedback. The deadline for reviewer-author discussion is approaching soon. We have put in tremendous effort in compiling a detailed response trying our very best to address all your concerns (please see the revised manuscript PDF and our P2P responses). If you are convinced and happy with our responses, please kindly consider re-evaluating/raising your final score; please also let us know if you have any further questions or concerns; we'll be more than happy to address those.
>
> Many thanks for your insightful feedback.
>
> Best regards, Authors.

---

> > ### Author Response · Authors · 2024-11-29
> > **Kind request for feedback**
> >
> > Dear Reviewer dDiR,
> >
> > Thank you for your thoughtful feedback on our paper. We kindly request that you confirm whether our responses have adequately addressed your questions. If there are any additional questions remaining, please do not hesitate to let us know!
> >
> > Additionally, if our rebuttal has addressed your comments, we would be most grateful if you could consider updating your scores to reflect that.
> >
> > We sincerely value your time and consideration and look forward to your feedback.
> >
> > Thank you once again!  Authors

---

> > > ### Comment · Reviewer_dDiR · 2024-12-01
> > > **Thank you for the response**
> > >
> > > I thank the authors for additional evaluations including fine-tuning on "unknown" samples. However, similar to other reviewers I still have my concerns on what is considered unknown to the diffusion model. I hence increase my score by 1 point signalling acceptance while also look forward to discuss with other reviewers during the reviewer's discussion phase.

---

> ### Author Response · Authors · 2024-12-02
> **Thanks!**
>
> We are happy that our responses have convinced you. We sincerely thank you for raising your score!
>
> However, we would like to further understand if any of the other reviewers share your concerns on the "unknown" classes. To the best of our understanding, we notice a similar but not exactly the same question raised by reviewer **X612** around potential data-leakage from the LAION pre-training set. To this aim, we have presented a global response to you and reviewer **X612**, addressing the applicability of GeNIe on unknown classes and clarifying our stance around potential data-leakage from the LAION dataset.
>
> Could you please check this global response and let us know if we have sufficiently addressed all your concerns? If not, we'd be happy to address them in the course of the time remaining for the discussion period.

---

### Official Review · Reviewer_X612 · 2024-11-04

**Soundness:** 3
**Presentation:** 3
**Contribution:** 2
**Rating:** 5
**Confidence:** 4

**Summary:**

This paper introduces a novel data augmentation method based on a latent diffusion model. It combines target and source categories in the latent noise-level to generate hard-negative samples for the source category. The impact of the proposed GeNIe is evaluated on two famous data-scarce problems.

**Strengths:**

- This paper proposes a realistic solution for practical problems of data augmentation using recent text-to-image generative models.
- The proposed method, GeNIe, contains an effective and adaptive strategy for data augmentation.

**Weaknesses:**

- The exact method for GeNIe is not detailed in the main script, although the key idea is presented, which can cause confusion.
- The proposed GeNIe is a simple data augmentation method on the latent space of Stable Diffusion, which lacks the novelty of the method except for modifying noise levels for better augmentation.
- GeNIe-Ada requires an additional classifier to search the decision boundary between two categories.

**Questions:**

- Is the classifier used for GeNIe-Ada the same as the one trained for the target task? How does the performance of this classifier affect the effectiveness of GeNIe-Ada?
- Is there no information leakage by using stable diffusion in few-shot classification for mini-ImageNet and tiered-ImageNet?
- Does using images from all other classes as the source image in line 352 mean that source images are selected from the support set of the other 4 classes, or all other 63 classes (for mini-ImageNet)?

---

> ### Author Response · Authors · 2024-11-22
> **Response to reviewer X612**
>
> > **[R1,W1] The exact method for GeNIe is not detailed in the main script**:
>
> Thanks for suggesting this. Barebone GeNIe is pretty simple. It uses a base Stable Diffusion (SD) model, takes as input a source image, selects a partial denosing level ($r$), which is optimized and automated by $\texttt{GeNIe-Ada}$, and a prompt to a different target class than that of the source image. This enables GeNIe to generate optimal hard-negative augmentations per source image and target class. We elabore details of our method in Lines 182-205. Following yor suggestion, we have now added a Pytorch-style pseudocode for GeNIe in Section A.5, Algorithm 2 of the appendix of our revised manuscript.
>
>
> ---
>
> > **[R1,W2] The proposed GeNIe is a simple data augmentation method on the latent space of Stable Diffusion, which lacks the novelty of the method except for modifying noise levels for better augmentation.**:
>
> We respectfully disagree with the reviewer. GeNIe is a simple, yet novel, generative data augmentation approach that uniquely leverages a diffusion-based models to combine two distinct sources of information—a source image and a contradictory target prompt—to generate hard-negative images. To the best of our knowledge, no other generative data augmentation method specifically explores the effectiveness of generative models in creating _hard-negative_ samples. Building upon GeNIe, we introduce a novel method (GeNIe-Ada) to effectively sample the optimal noise level, eliminating the need for manual parameter tuning and further enhancing GeNIe's performance. While we acknowledge that GeNIe is a straightforward approach (unlike the more complex GeNIe-Ada), it demonstrates strong effectiveness across a wide range of benchmarks, as highlighted in our work.
>
> ---
>
> > **[R1,W3] GeNIe-Ada requires an additional classifier to search the decision boundary between two categories.**:
>
> Thanks for the comment. We would like to emphasize that GeNIe-Ada only requires the downstream task's feature extractor/backbone ($f_{\theta}$), and not its classifier. We have clarified this in the revised manuscript. The backbone is used to extract the image-embeddings $Z_r$ corresponding to their partially-noised images $X_r = \texttt{STDiff}(X_S, P, r)$, as illustrated in Algorithm 1 and Section 3. This backbone is pre-trained on the few-shot training set and is thus already available during few-shot testing. This does not introduce any additional requirements specific to our generative hard-negative augmentation method.

---

> ### Author Response · Authors · 2024-11-22
> **Response to reviewer X612**
>
> > **[R1,Q1] Is the classifier used for GeNIe-Ada the same as the one trained for the target task? How does the performance of this classifier affect the effectiveness of GeNIe-Ada?**:
>
> Thanks for the insightful question which prompted the following experiment. Yes, in all experiments presented in this paper, we use the same backbone for $f_{\theta}(.)$ that is subsequently fine-tuned for few-shot classification tasks. However, to analyze the effect of the backbone feature extractor $f_{\theta}$ on selecting the optimal hard-negative using GeNIe-Ada, we use an IN-1k pre-trained DeiT base instead of the UniSiam pretrained ResNet backbone. However, we still utilize the same ResNet backbone for few-shot classification. We have now included Table A2 in the appendix which demonstrates the results of this experiment. We notice a marginal improvement of upto 0.7% when using GeNIe-Ada+DeiT-B as compared to GeNIe-Ada, which uses the UniSiam pre-trained ResNet backbone. This suggests that there is still potential to develop more effective strategies for selecting noise ratios to further enhance GeNIe. However, in this paper, we limit our exploration to GeNIe-Ada and leave these improvements for future work.
>
> ---
>
> ### ResNet-18
> | **Augmentation** | **Noise Ratio Selector Backbone** | **Method [Classifier Backbone]** | **1-shot** | **5-shot** |
> |------------------|----------------------------------|-----------------------------------|------------|------------|
> | **GENIE (Ours)** | -                                | UniSiam [ResNet18]               | **75.5±0.6** | **85.4±0.4** |
> | **GENIE-Ada**  | UniSiam [ResNet18]               | UniSiam [ResNet18]               | **76.8±0.6** | **85.9±0.4** |
> | **GENIE-Ada**  | IN-1K [DeiT-B]                   | UniSiam [ResNet18]               | **77.5±0.5** | **86.3±0.2** |
>
> ---
>
> ### ResNet-50
> | **Augmentation** | **Noise Ratio Selector Backbone** | **Method [Classifier Backbone]** | **1-shot** | **5-shot** |
> |------------------|----------------------------------|-----------------------------------|------------|------------|
> | **GENIE**        | -                                | UniSiam [ResNet50]               | **77.3±0.6** | **87.2±0.4** |
> | **GENIE-Ada**  | UniSiam [ResNet50]               | UniSiam [ResNet50]               | **78.6±0.6** | **87.9±0.4** |
> | **GENIE-Ada**  | IN-1K [DeiT-B]                   | UniSiam [ResNet50]               | **79.2±0.4** | **88.3±0.5** |
>
>
>
> ---
>
> > **[R1,Q2] Is there no information leakage by using stable diffusion in few-shot classification for mini-ImageNet and tiered-ImageNet?**:
>
> This a great comment. To substantiate our understanding per your remark, we have run a set of experiments on the few-shot setting as summarized in Fig. A3 of the appendix. To set the scene, we use pretrained image encoder as an oracle (DeiT-Base) to extract the latent embeddings corresponding to the train (i.e. support) set, test (i.e. query set) and augmentations generated by GeNIe. Fig. A3 in the revised manuscript demonstrates the distribution of distances between train-test and augmentation-test pairs across 600 episodes. As can be seen, the (mean of the) distribution of augmentation-test pair is higher that that of train-test pair indicating that the augmented samples are indeed different from the test sets (based on the strong assumption of train and test sets being mutually exclusive). This is further illustrated in the last column of Fig. A3 on a UMAP embedding plot of a random episode where the embedding of train, test and augmentations are plotted. Here again there is noticeable separation between the augmentation and test samples as compared to train and test samples.
>
> ---
>
> > **[R1,Q3] Does using images from all other classes as the source image in line 352 mean that source images are selected from the support set of the other 4 classes, or all other 63 classes (for mini-ImageNet)?**:
>
> We use the source images from the support set of the other 4 classes for augmentation in a given episode, and not from all the other 63 classes, thus preventing leakage of data across episodes.

---

> ### Author Response · Authors · 2024-11-26
> **Follow Up - Deadline Approaching**
>
> Dear Reviewer X612,
>
> Firstly, many thanks for your rigorous and constructive feedback. The deadline for reviewer-author discussion is approaching soon. We have put in tremendous effort in compiling a detailed response trying our very best to address all your concerns (please see the revised manuscript PDF and our P2P responses). If you are convinced and happy with our responses, please kindly consider re-evaluating/raising your final score; please also let us know if you have any further questions or concerns; we'll be more than happy to address those.
>
> Many thanks for your insightful feedback.
>
> Best regards, Authors.

---

> > ### Author Response · Authors · 2024-11-29
> > **Kind request for feedback**
> >
> > Dear Reviewer X612,
> >
> > Thank you for your thoughtful feedback on our paper. We kindly request that you confirm whether our responses have adequately addressed your questions. If there are any additional questions remaining, please do not hesitate to let us know!
> >
> > Additionally, if our rebuttal has addressed your comments, we would be most grateful if you could consider updating your scores to reflect that.
> >
> > We sincerely value your time and consideration and look forward to your feedback.
> >
> > Thank you once again!
> > Authors

---

> > > ### Comment · Reviewer_X612 · 2024-11-29
> > >
> > > Thank you for your response, and I have some follow-up questions and remaining concerns.
> > >
> > > 1. GeNIe-Ada uses an auxiliary classifier backbone to find the proper noise level that transitions from the source category to the target category occur. It seems more soundness for me to use VAE encoder used for Stable Diffusion in this case to get the latents. Are there any rationales for using an auxiliary classifier?
> > >
> > > 2. I still have a concern about information leakage, and responses from the authors do not address it for my side. My concern is about using large generative models (Stable Diffusion) whose training set could contain the contents of target datasets (ImageNet). As long as I know, it is not clear whether curated LAION dataset for SD 1.5 contains ImageNet or its variants.
> > >
> > > I appreciate again for your supportable responses.

---

> ### Author Response · Authors · 2024-12-01
> **Response to reviewer X612**
>
> > **[R1,Q1] GeNIe-Ada uses an auxiliary classifier backbone to find the proper noise level that transitions from the source category to the target category occur. It seems more soundness for me to use VAE encoder used for Stable Diffusion in this case to get the latents. Are there any rationales for using an auxiliary classifier?**
>
> That is a great remark. The rationale behind using the donwstream (few-shot) task pre-trained backbone is to utilize its class-aware/relevant latent space. On the other hand, the VAE used in SDv1.5 is pre-trained using an image-reconstruction task, which is not the most relevant for embedding class-cognizant information in the latent space. For eg: UniSiam uses a mutual-information maximization loss between images and their augmentations, thus allowing the representations to be instance-discriminative/class-aware. Also, since this backbone is anyway used for test-set few-shot classification, we re-use this backbone for GeNIe-Ada to avoid introducing any additional resource requirements.
>
> Following your suggestion, we empirically evaluate the use of VAE latent space for GeNIe-Ada. We conduct an experiment where optimal samples in GeNIe-Ada are selected using the latent space of the VAE encoder used in SDv1.5. As illustrated in the table below, the few-shot accuracies of GeNIe-Ada with a VAE noise selector backbone reduce as compared to the version with the UniSiam pre-trained backbone. We attribute this drop in scores to the inability of the VAE backbone in identifying meaningful class-relevant semantic information present in the generated images. We attribute this to the inherent nature of VAE's pre-training task - image reconstruction. On the other hand, the UniSiam pre-trained backbone involves discriminative pre-training, a task that is more relevant and apt to the downstream task of image discrimination/classification. Additionally, since we already have access to this pre-trained backbone for few-shot image classification, this is not considered as an additional resource requirement.
>
>
>
> ### ResNet-18
> | **Augmentation** | **Noise Ratio Selector Backbone** | **Method [Classifier Backbone]** | **1-shot** | **5-shot** |
> |------------------|----------------------------------|-----------------------------------|------------|------------|
> | **GeNIe** | -                                | UniSiam [ResNet18]               | **75.5±0.6** | **85.4±0.4** |
> | **GeNIe-Ada**  | UniSiam [ResNet18]               | UniSiam [ResNet18]               | **76.8±0.6** | **85.9±0.4** |
> | **GeNIe-Ada**  | VAE - SDv1.5                  | UniSiam [ResNet18]               | **68.8±0.7** | **78.9±0.8** |
>
> ---
>
> ### ResNet-50
> | **Augmentation** | **Noise Ratio Selector Backbone** | **Method [Classifier Backbone]** | **1-shot** | **5-shot** |
> |------------------|----------------------------------|-----------------------------------|------------|------------|
> | **GeNIe**        | -                                | UniSiam [ResNet50]               | **77.3±0.6** | **87.2±0.4** |
> | **GeNIe-Ada**  | UniSiam [ResNet50]               | UniSiam [ResNet50]               | **78.6±0.6** | **87.9±0.4** |
> | **GeNIe-Ada**  | VAE - SDv1.5                   | UniSiam [ResNet50]               | **70.7±0.6** | **81.5±0.7** |

---

> > ### Author Response · Authors · 2024-12-01
> > **Response to reviewer X612**
> >
> > > **[R1,Q2] I still have a concern about information leakage, and responses from the authors do not address it for my side. My concern is about using large generative models (Stable Diffusion) whose training set could contain the contents of target datasets (ImageNet). As long as I know, it is not clear whether curated LAION dataset for SD 1.5 contains ImageNet or its variants.**
> >
> >
> > - Thanks for the constructive remark. We would like to clarify that the goal of our experimental study illustrated in Fig. A3 is to empirically demonstrate that GeNIe, GeNIe-Ada _do not_ create augmentations that mimic the few-shot test-set images of miniImagenet. This is illustrated by the by the noticeable separation between the GeNIe augmentations and test samples as compared to train and test samples, in both, the UMAP embedding plot and the distance-distribution plots across all $r \in [0.5, 0.8]$ in Fig. A3.
> >
> > - Since LAION is a large-scale (400 million samples), noisy, internet-sourced dataset, there is a possibility that it encapsulates images common in several downstream datasets. In such cases, where categories are known to the LAION pre-trained diffusion model, **the text-to-image (Txt2Img) baseline generates images/augmentations that closely resemble the few-shot test-set samples**. **GeNIe and GeNIe-Ada go beyond this naïve methodology** of generating test-set similar augmentations by finding the optimal hard-negatives per sample per target-class, that are not only different from the test-set (as indicated in Fig. A3), but are also significantly more effective in improving few-shot classification performance (as indicated in Tables 1,2,3,4).
> >
> > - All in all, GeNIe and GeNIe-Ada provide a _simple yet effective_ way of utlizing large-scale pre-trained generative foundation models, such as SDv1.5, for data-deficient image classification tasks. While Txt2Img offers a simple approach to augment datasets for downstream tasks, our paper builds on this idea by finding optimal hard-negative samples to enhance downstream task performance, an unexplored concept in the literature.
> >
> > - Additionally, for scenarios where target categories are unknown or absent in LAION, we propose fine-tuning target embeddings with textual inversion, followed by GeNIe for generating hard-negative images. Results in Section A6 show GeNIe surpassing the baseline (DAFusion), which relies solely on fine-tuned embeddings, further validating the effectiveness of hard negatives in GeNIe.
> >
> > **Remark:**
> >
> > Does our response address all your concerns?
> >
> > We would be happy to provide any further clarifications if you have any more questions or concerns. If not, we would greatly appreciate it if you would re-evaluate our paper's score to give our work a chance to benefit the community from its acceptance.

---

> > > ### Author Response · Authors · 2024-12-02
> > > **Summary of responses and request for feedback**
> > >
> > > Dear Reviewer **X612**,
> > >
> > > We thank you for your insightful remarks and constructive suggestions for additional experiments. As the rebuttal/discussion period is coming to an end, we would like to summarize our responses to your remarks and the new experimental evaluations we carried out based on your feedback.
> > >
> > > 1. We address your remark of not describing the exact methodology of GeNIe in the manuscript, by now including a pytorch-style pseudocode of GeNIe in Algorithm A2, Section A5 of the revised draft.
> > > 2. We address your constructive remark on the novelty and contributions of GeNIe, GeNIe-Ada by now clearly highlighting that GeNIe's novelty lies in its unique ability to leverage a diffusion-model generating _hard-negative augmentations_.
> > > 3. To answer your question regarding the backbone's effect on the effectiveness of GeNIe-Ada, we conducted an experiment with a DEiT-Base backbone to select the optimal hard-negatives with GeNIe-Ada (included in Table A2, Section A4). Furthermore, to clarify your concerns on using the SDv1.5 VAE as the backbone, we conducted an additional experiment with the VAE backbone for optimal noise selection.
> > > 4. To address your remark on information leakage between LAION and mini/tieredImagenet, we first presented statistical and qualitative results on image embeddings ensuring discrepancy between the generated augmentations and the test set in Fig. A3 of the revised manuscript. To further clarify on your concerns of data leakage, we accentuated on the fact that GeNIe goes beyond the naive methodology of generating LAION/few-shot test-set similar augmentations (as is the case in Txt2Img), by generating optimal hard-negatives for enhancing performance.
> > >
> > >
> > > Furthermore, we want to draw your attention to a set of additional experiments we have conducted during the discussion period for the other reviewers, that have helped us improve the quality of the (updated) revised manuscript.
> > >
> > > 1. In response to reviewer **dDiR** and **ny5s**, we demonstrate new experimental results with: (i.) a Mixed augmentation strategy, (ii.) GeNIe alongside the MixUp augmentation strategy and (iii.) Combinations of Mixed+CutMix and Mixed+MixUp augmentation strategies.
> > > 2. To further answer reviewer **ny5s**'s questions, we conducted additional empirical studies of GeNIe's semantic transitioning between source and target classes using DINOv2 representations in Fig. A5, A6. We also put together a new table that compares the performance of recent (2022, 2023) SoTa few-shot learning methods against GeNIe and GeNIe-Ada augmentations. These additional experiments led to an increase in reviewer **ny5s**'s score to 8.
> > > 3. In response to reviewer **eEnJ** and **dDiR**'s concerns on limited evaluations and GeNIe's applicability to cases where target classes are unknown to the diffusion model, we demonstrate the robustness of GeNIe in these cases by parameter-efficient fine-tuning the base diffusion model on few images of the target class using Textual-Inversion. This has been included in Table A3, Section A6 of the appendix of our revised manuscript. This led to an increase in reviewer **dDiR**'s score to 6.
> > >
> > >
> > > We would be more than happy to provide further clarifications and revisions if you have any more questions or concerns, and if not we would greatly appreciate it if you would please re-evaluate our paper's score. Thank you again for your feedback which helped us tremendously to improve our paper!

---

> > > > ### Author Response · Authors · 2024-12-03
> > > > **Kind request for score re-evaluation**
> > > >
> > > > Thanks you all for your insightful remarks and constructive suggestions for additional experiments that have helped us further improve our work!
> > > >
> > > > We have put in tremendous efforts to answer all your questions and clarify your concerns, as summarized in the response above and in the general comment "General Response to reviewers dDiR and X612". If you feel we have addressed all your remarks, we would greatly appreciate it if you would please re-evaluate our paper's score to give our work a chance to benefit the community from its acceptance.

---

### Author Response · Authors · 2024-11-22
**General Response to ALL reviewers**

We do appreciate reviewer's constructive feedback which helped to further improve the quality and clarity of the paper. We are please by the positive feedback from reviewer **X612** for finding our proposed method "effective" and "a realistic solution for practical problems", from reviewer **dDiR** for finding our paper "well-written and easy to follow", and for finding our motivation behind GeNIe "sound". We appricate reviewer **eEnJ**'s comment on GeNIe-Ada as an extension that "adds significant value", and reviewer **ny5s**'s comment on our "extensive" experimental evaluation.

After perusing reviewer's remarks and recommendations, we have put in tremendous effort to provide further evidence (new experimentation and qualitative demonstrations) to corroborate the efficacy of GeNIe as summarized below:
- In response to reviewer **X612** and **dDiR**, regarding information leakage between LAION and mini/tieredImagenet, we present a statistical results on image embeddings ensuring discrepancy between the generated augmentations and the test set;
- In response to reviewer **X612**, we present new few-shot classification results with a different backbone for selecting optimal hard-negatives in the case of GeNIe-Ada.
- In response to reviewer **dDiR**, we demonstrate the robustness of GeNIe in cases where the target class is unknown to the diffusion model by parameter-efficient fine-tuning the base diffusion model on few images of the target class.
- In response to reviewer **eEnJ**, We include additional experiments to further demonstrate the effectiveness of GeNIe in scenarios where category labels are unavailable and compare its performance with the DAFusion baseline.
- In response to reviewer **dDiR**, **ny5s**, we demonstrate new experimental results with: (i.) a Mixed augmentation strategy, (ii.) GeNIe alongside the MixUp augmentation strategy and (iii.) Combinations of Mixed+CutMix and Mixed+MixUp augmentation strategies.

**Remark:** Note that the additional experimental results and explanations have been added to the revised manuscript in text color **blue**.

We do hope this addresses reviewer's concerns and questions, and look forward to engaging further during reviewer-author discussion period.

---

### Author Response · Authors · 2024-12-02
**General Response to reviewers dDiR and X612 - Part 1**

### General Response to reviewers **dDiR** and **X612**:

We do appreciate your constructive feedback which helped to further improve the quality and clarity of our paper. We notice a common theme in your final remarks around (i). GeNIe's potential limited applicability to tasks with SDv1.5 "known" classes, and (ii). Potential data overlap between LAION and few-shot test sets. To answer these two, we have presented new experimental results and further clarifications around our method:

1. GeNIe generates optimal hard-negative samples using two sources of information - i. a source image, and ii. a target class. In a challenging scenario where the target classes are "unknown" to the diffusion model, GeNIe can still utilize other _distractor_ classes that do not belong to the set of the unknown target clasess. These distractor classes can be selected from a set of top-k other categorizes different from the source class yet similar to it with respect to their CLIP text-embeddings. As an extension to our work, we could also utilize an LLM such as GPT-4 to query a target class that might be confused with the source image's class, to be used as the text-prompt for the diffusion model.

2. To demonstrate that GeNIe is still applicable in cases where the target class is not known to the diffusion model, we test its performance in a scenario where we _do not_ input target classes as text-prompts into the diffusion model. Instead, we now use textual inversion to fine-tune the diffusion model (embeddings) on a few images belonging the unknown target classes. This fine-tuning allows the diffusion model to learn embeddings specific to the target class, enabling the generation of the desired hard-negative examples. Note that once the diffusion model is fine-tuned, the procedure to generate hard-negatives using partial noising and a contradictory text-prompt remains the same. As can be seen in Table A3 (also presented below), GeNIe+TxtInv performs significantly better than the DAFusion baseline by offering gains in few-shot accuracy of up to 9%. We would like to re-iterate that in this case, we _do not utilize any information about the target category labels_. DAFusion also employs textual-inversion-based fine-tuning; however, it does so without generating hard-negative samples. GeNIe's superior performance indicates that it is effective even in scenarios where the diffusion model is _unaware of the target-class_.



### ResNet-18

| Augmentation       | Method          | Pre-training | 1-shot         | 5-shot         |
|--------------------|-----------------|--------------|----------------|----------------|
| DAFusion (2024)    | UniSiam (2023)  | unsup.       | 64.3 ± 1.8     | 82.0 ± 1.4     |
| **GeNIe+TxtInv**   | UniSiam (2023)  | unsup.       | **73.9 ± 0.8** | **84.6 ± 0.9** |

### ResNet-50

| Augmentation       | Method          | Pre-training | 1-shot         | 5-shot         |
|--------------------|-----------------|--------------|----------------|----------------|
| DAFusion (2024)    | UniSiam (2023)  | unsup.       | 65.7 ± 1.8     | 83.9 ± 1.2     |
| **GeNIe+TxtInv**   | UniSiam (2023)  | unsup.       | **76.2 ± 1.2** | **86.2 ± 0.9** |

---

> ### Author Response · Authors · 2024-12-02
> **General Response to reviewers dDiR and X612 - Part 2**
>
> ### General Response to reviewers **dDiR** and **X612**:
>
> 3. To address the comment on potential data overlap, we would like to clarify that Fig. A3 demonstrates that GeNIe, GeNIe-Ada _do not_ create augmentations that mimic the test-set images of miniImagenet. This is illustrated by the by the noticeable separation between the GeNIe augmentations and test samples as compared to train and test samples, in both, the UMAP embedding plot and the distance-distribution plots across all $r \in [0.5,0.8]$. Even if we do consider the case that LAION encapsulates images common in several downstream datasets, we empirically demonstrate that GeNIe and GeNIe-Ada go beyond the naïve methodology of generating test-set similar augmentations by generating hard-negatives that are not only different from the test-set (as indicated in Fig. A3), but are also significantly more effective in improving few-shot classification performance as compared to the Txt2Img baseline (as indicated in Tables 1,2,3,4).
>
> All in all, we demonstrate that GeNIe and GeNIe-Ada do not naively augment the test-set by copying the LAION pre-training dataset, while also being effectively applicable in scenarios where the target-class information is "unknown" to the diffusion-model.
>
> **Remark:**
>
> We do believe that we have presented necessary and sufficient empirical evidence to support and substantiate our claims around your remarks. However, one last possible direction to experiment on would be to further enhance our experimental evaluation by incorporating datasets such as ChestX[1] or EuroSAT[2], where the dataset classes are significantly more novel to the SDv1.5 model. However, this requires more time than the rebuttal period allows and thus, we would be happy to include our experimental evaluations on one of these datasets in the final version of our revised manuscript upon acceptance. Thanks again for your valuable and constructive feedback!
>
> **References:**
>
> [1] Xiaosong Wang, et al. Chestx-ray8: Hospital-scale chest x-ray database and benchmarks on weakly-supervised
> classification and localization of common thorax diseases. In Proceedings of the IEEE CVPR, pp. 2097–2106, 2017.
>
> [2] Patrick Helber, et al. Eurosat: A novel dataset and deep learning benchmark for land use and land cover classification. IEEE Journal of Selected Topics in Applied Earth Observations and Remote Sensing, 2019.

---

### Author Response · Authors · 2024-12-02
**Summary of the discussion period**

Dear reviewers and the AC,

We appreciate your constructive feedback which helped us further improve the quality and clarity of the paper. We are pleased by the positive feedback from the reviewers for finding our proposed method "effective", "novel yet elegantly simple", "a realistic solution for practical problems", and for finding our paper "well-written and easy to follow", and for finding our motivation behind GeNIe "sound".

As the discussion period comes to an end, we would like to summarize the tremendous efforts we have put in during the rebuttal period (new sets of experimental results and qualitative demonstrations) to address the reviewers’ remarks as follows:

- In response to reviewer **X612** and **dDiR**, regarding a potential distribution overlap between LAION and mini/tieredImagenet, we presented statistical and qualitative results on image embeddings ensuring discrepancy between the generated augmentations and the test set. These have been added to Fig. A3 in Section A2 of the revised draft.
- In response to reviewer **X612**, we presented new few-shot classification results with two different backbones - DEit-B and VAE of SDv1.5, for selecting optimal hard-negatives in the case of GeNIe-Ada. These have been added to Table A2 in Section A4 of the revised draft. Additionally, we have also added a detailed pytorch-style pseudocode of GeNIe in Alg.2 Section A5 of the revised draft.
- In response to reviewer **dDiR**, we demonstrated the robustness of GeNIe in cases where the target class is unknown to the diffusion model by parameter-efficient (textual-inversion) fine-tuning the base diffusion model on a few images of the target class and demonstrate superior performance than the DAFusion baseline, on-par performance with GeNIe, GeNIe-Ada. These have been added to Table A3 in Section A6 of the revised draft.
- In response to reviewer **dDiR**, **ny5s**, we shared new experimental results with: (i.) a Mixed augmentation strategy, (ii.) GeNIe alongside the MixUp augmentation strategy and (iii.) Combinations of Mixed+CutMix and Mixed+MixUp augmentation strategies. These are now included in Table A1, Section A3 of our revised draft.
- In follow-up responses to reviewer **ny5s**, we demonstrated a new qualitative analysis of 10 different semantic trajectories of GeNIe with the same source and target classes. We observe that each trajectory has a separate rate/speed of semantic shift from source to target, while following the same path. These are included in Fig A5, Section A12 of the revised draft. Additionally, we provide further insight into GeNIe's category transitions by illustrating generated images across different $r$'s in which GeNIe was applied twice to transform an image between a source image and a target category. These have now been added to Fig. A6, Section A14 of the revised draft.

Our responses detailed above have led to an increase in reviewer **ny5s**'s score to 8 and reviewer **dDiR**'s score to 6.

**Remark:** Note that the additional experimental results and explanations have been added to the revised manuscript in $\textcolor{blue}{blue}$.

We do hope our additional experiments and responses have addressed all the reviewers' concerns and questions. If so, we would greatly appreciate it if you please kindly consider re-evaluating/increasing our paper's final score.

---

### Meta-Review · Area_Chair_s2GD · 2024-12-19

**Metareview:**

This paper proposes to leverage generative models to obtain data augmentations that improve downstream performance of few shot and long tail datasets. Augmentations are produced by combining a source image with a target category prompt.

The paper was reviewed by four knowledgeable reviewers who acknowledged that the paper is well written and easy to follow (dDiR), the motivation sound (dDiR), and the solution relatively effective (X612). The reviewers raised concerns about:

1. Missing information on the method (X612)
2. Limited novelty (eEnJ), strong assumptions made (X612, dDiR)
3. Potential information leakage (from pre-trained generative models and about ImageNet) (X612)
4. Missing some related work discussion/comparisons (eEnJ, dDiR)
5. Possible inefficiency of the approach (ny5S)
6. Unclear significance of results (e.g. unclear whether improvements come from the method or from the pre-trained diffusion model) (dDiR).

During rebuttal/discussion, the authors discussed the questions related to SDv1.5 known classes, and potential data overlap between pre-training datasets and few-shot test set. They provided missing details on the method and experimental setup, argued for the novelty of their approach, shared additional experiments analyzing the effect of the feature extractor, discussed the surfaced leakage issue, added some comparisons suggested by the reviewers, as well as combining the proposed approach with standard data augmentation techniques, and promised results on additional datasets. After rebuttal, one reviewer signals clear accept, one leans towards rejection, and the remaining two appear to position themselves as borderline. However, as pointed by the authors, some reviewers remain unresponsive during discussion time. Given this, the AC took the time to go over the paper, the reviews and discussions. Although the AC agrees with the reviewers that the proposed method may hold potential, the AC remains unconvinced by the experimental evidence provided. In particular, although prior art is discussed in the related work, important comparisons are missing in the experimental section (e.g. StableRep, feedback guidance, and Fill Up). Compared to Feedback Guidance and Fill Up, the improvements achieved by Genie on the Few split of the ImageNet-LT dataset, appear less impressive. It is also unclear what the effect of the number of synthetic images is on the downstream performance, or whether using a more recent text-to-image model would result in different findings. For these reasons, and the remaining concerns of the reviewers, the AC cannot recommend acceptance, despite the merits outlined by some reviewers. The AC encourages the authors to consider the feedback received to improve future iterations of their work.

**Additional Comments On Reviewer Discussion:**

See above.

---

### Decision · Program_Chairs · 2025-01-22

Reject